# REVE: A Foundation Model for EEG Adapting to Any Setup with Large-Scale Pretraining on 25,000 Subjects

Yassine El Ouahidi[1],[*] Jonathan Lys[1], Philipp Thölke[2], Nicolas Farrugia[1],
Bastien Pasdeloup[1], Vincent Gripon[1], Karim Jerbi[2,3,4], Giulia Lioi[1*]

[1] IMT Atlantique, Lab-STICC, UMR CNRS 6285, F-29238 Brest, France
[2] Psychology Department, Université de Montréal, Montreal, QC, Canada
[3] Mila (Quebec AI research institute), Montreal, QC, Canada
[4] UNIQUE (Quebec Neuro-AI research center), QC, Canada

## Abstract

Foundation models have transformed AI by reducing reliance on task-specific data through large-scale pretraining. While successful in language and vision, their adoption in EEG has lagged due to the heterogeneity of public datasets, which are collected under varying protocols, devices, and electrode configurations. Existing EEG foundation models struggle to generalize across these variations, often restricting pretraining to a single setup, resulting in suboptimal performance, in particular under linear probing. We present REVE (Representation for EEG with Versatile Embeddings), a pretrained model explicitly designed to generalize across diverse EEG signals. REVE introduces a novel 4D positional encoding scheme that enables it to process signals of arbitrary length and electrode arrangement. Using a masked autoencoding objective, we pretrain REVE on over 60,000 hours of EEG data from 92 datasets spanning 25,000 subjects, representing the largest EEG pretraining effort to date. REVE achieves state-of-the-art results on 10 downstream EEG tasks, including motor imagery classification, seizure detection, sleep staging, cognitive load estimation, and emotion recognition. With little to no fine-tuning, it demonstrates strong generalization, and nuanced spatio-temporal modeling. We release code, pretrained weights, and tutorials[2] to support standardized EEG research and accelerate progress in clinical neuroscience.

## 1 Introduction

Electroencephalography (EEG) is a non-invasive technique widely used to study brain activity, with applications spanning brain-computer interfaces (BCIs), clinical diagnostics, and neuroscience research. Despite its potential, the adoption of EEG-based technologies remains limited (Lotte et al., 2018). A key challenge is developing models that generalize effectively to new subjects. EEG data varies widely in electrode configurations, recording conditions, and subject-specific factors, complicating model transferability. This heterogeneity has led to a fragmented ecosystem of datasets and task-specific models, many of which struggle to generalize across settings.

Foundation models have transformed natural language processing (Achiam et al., 2023; Dubey et al., 2024; Warner et al., 2024) and computer vision (Radford et al., 2021; Caron et al., 2021; Kirillov et al., 2023) by leveraging large-scale pretraining to enable transfer with minimal supervision. Their

---

[*]Corresponding authors: yassine.elouahidi@mistral.ai, giulia.lioi@imt-atlantique.fr
[2]Project page: https://brain-bzh.github.io/reve/

39th Conference on Neural Information Processing Systems (NeurIPS 2025).

ability to produce general-purpose representations has sparked growing interest in building similar models for EEG (Yang et al., 2024; Wang et al., 2024b; Jiang et al., 2024; Cui et al., 2024; Yuan et al., 2024b; Wang et al., 2024a). Yet, EEG poses unique challenges including data heterogeneity, low signal-to-noise ratio, and the lack of standardized positional encoding to accommodate varying electrode configurations.

Recent EEG foundation models such as BIOT (Yang et al., 2024), Labram (Jiang et al., 2024), CBraMod (Wang et al., 2024b), and NeuroGPT (Cui et al., 2024) adopt self-supervised learning (SSL) techniques for pretraining. While promising, many of these models rely solely on the TUH database (Obeid and Picone, 2016) which uses a fixed 19 or 21-channel montage. As a result, they often fail to generalize to datasets with different electrode layouts or recording setups. Furthermore, existing positional encoding schemes, whether absolute (Yang et al., 2024; Jiang et al., 2024) or convolutional (Wang et al., 2024b), lack the flexibility to accommodate spatial diversity, often necessitating full fine-tuning for transfer.

To address the limitations in current EEG foundation models, we consider three core contributions that enable scalable, generalizable representation learning across diverse, large-scale EEG datasets.

First, we propose a novel 4D positional encoding scheme that enables flexible modeling of EEG signals with varying temporal lengths and electrode configurations. Unlike existing absolute or convolutional encodings, our formulation naturally supports spatial and temporal variability, eliminating the need for fixed montages or fine-tuning of positional priors.

Thanks to this flexible positional encoding method, we are able to train with a wider range of EEG configurations, allowing to scale to larger and more heterogeneous datasets. To this end, we curate the largest and most diverse EEG corpus to date, comprising over 60,000 hours of data from 92 datasets and 25,000 subjects. This diverse collection spans clinical, BCI, and research domains, providing the scale and diversity necessary for robust pretraining.

Combining architectural flexibility with large-scale data results in REVE (Representation for EEG with Versatile Embeddings), a spatio-temporal transformer model trained with a modified masked autoencoder (MAE) (He et al., 2022) objective that promotes learning better representations in the model. REVE learns general-purpose EEG representations that transfer effectively across a wide range of downstream tasks.

REVE achieves state-of-the-art performance across numerous benchmarks, including BCI and clinical datasets, outperforming prior EEG foundation models. Our scaling studies further show improved generalization with larger model sizes, reinforcing the benefits of large-scale pretraining. To support adoption, we release open-source code, pretrained models of multiple sizes, and detailed tutorials for applying REVE to various EEG tasks. By addressing the unique challenges of EEG with scalable architectures and flexible spatial encoding, REVE establishes a unified foundation for EEG analysis and paves the way for new advances in neuroscience and clinical applications.

## 2 Methods

We pretrain our encoder using a masked autoencoder objective. The REVE encoder consists of a patch embedding module, a 4D position encoding module, and a transformer backbone. During pretraining, we apply spatio-temporal contiguous masking to the patch embeddings and jointly train the encoder and decoder to reconstruct the missing segments of EEG, enabling the encoder to learn robust feature representations. Subsequent hyperparameter values are listed in Table 5 in the Appendix.

### 2.1 EEG Representation and Block Masking strategy

We represent multi-channel EEG data as $\mathbf{X} \in \mathbb{R}^{C \times T}$, where $C$ is the number of electrodes and $T$ the number of time samples, electrode positions are given by $\mathbf{P} \in \mathbb{R}^{C \times 3}$, corresponding to their 3D coordinates. To process the data, we segment each channel into patches of size $w$ with overlap $o$, following BIOT (Yang et al., 2024). This yields $p = \left\lceil \frac{T-w}{w-o} \right\rceil + \mathbb{1}\left[(T-w) \bmod (w-o) \neq 0\right]$ non-overlapping patches (discarding any incomplete ones), and reshapes $\mathbf{X}$ into $\mathbf{Xp} \in \mathbb{R}^{C \times p \times w}$. Each patch is linearly embedded, resulting in $\mathbf{E} \in \mathbb{R}^{C \times p \times D_E}$, where $D_E$ is the embedding dimension.

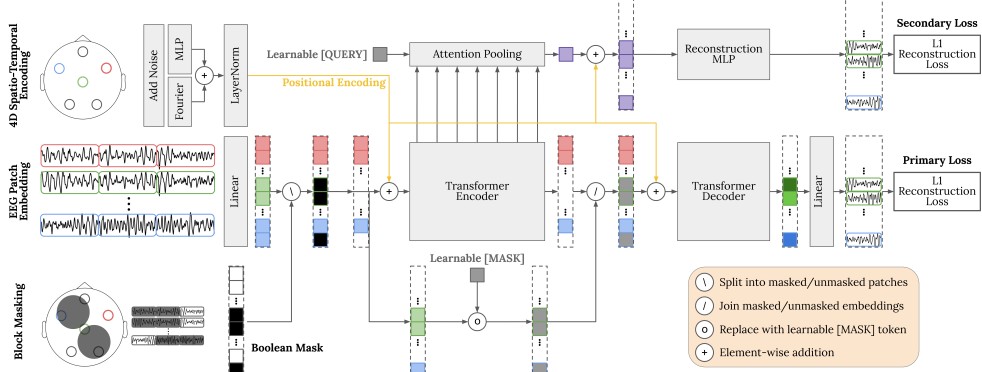

Figure 1: Overview of the **REVE** pretraining framework. The model processes multi-channel EEG data through a linear **Patch Embedding** where signals are divided into overlapping temporal patches for each channel and embedded with a linear layer. **4D Spatio-Temporal Position Encoding** combines spatial coordinates of electrodes with temporal patch indices, augmented with noise for robust generalization. A **Block Masking Strategy** masks contiguous regions across spatial and temporal dimensions to simulate realistic disruptions. The transformer encoder processes unmasked embeddings. Updated embeddings are joined with learnable placeholders for the masked tokens, from which raw EEG is reconstructed using the decoder. The **Primary Task** predicts raw EEG signals directly, while the **Secondary Task** trains a single global token via attention pooling to summarize the input. Both tasks minimize an $L_1$ reconstruction loss.

To enhance learning during pretraining, we apply a joint spatio-temporal block masking strategy that masks structured regions across both spatial and temporal dimensions. Random masking, proposed for EEG by Chien et al. (2022), was later improved through spatial masking (Mohammadi Foumani et al., 2024; Guetschel et al., 2024). In this work, we extend the masking strategy to the temporal domain. This builds on insights from image modeling, where structured masking outperforms random masking (Xie et al., 2022), a trend also supported by our ablation results (Table 18, Appendix). As neighboring segments of EEG, in both spatial and temporal domain, are typically similar, naive random masking could leave redundant information exposed, reducing the difficulty of reconstruction. In contrast, block masking better disrupts these patterns, encouraging more effective learning.

Our block masking strategy is governed by the following parameters: The masking Ratio $M_r$ controls the overall proportion of masked tokens. The spatial Masking Radius $R_s$ and Temporal Masking Radius $R_t$ respectively define the spatial extent (around a selected channel) and the time window (around a selected token) to be masked. Similarly, the Dropout Ratio $D_r$ sets the fraction of masked tokens for which the entire time series of the corresponding channel is dropped, while the dropout Radius $R_d$ determines the spatial neighborhood affected by dropout. For tokens not dropped, temporal masking is applied within radius $R_t$. This process yields a binary mask $\mathbf{B} \in \mathbb{R}^{C \times p}$, containing $N_{\mathrm{m}} = \lfloor (1 - M_r) \cdot C \cdot p \rfloor$ masked entries (zeros) and $N_{\bar{\mathrm{m}}} = C \cdot p - N_{\mathrm{m}}$ unmasked entries (ones).

## 2.2   4D Position Encoding Strategy

Unlike prior works that rely on learned embedding tables for spatial encoding vectors (Jiang et al., 2024; Wang et al., 2024b), we directly generate position encodings from the spatio-temporal coordinates of the tokens, allowing the processing of signals of any length or EEG layout and enabling better generalization to unseen setups. More specifically, our method uses a transformation applicable to each position, utilizing the actual 3D coordinates and timestep of each EEG patch, enabling the model to handle arbitrary electrode configurations and sequence lengths without relying on learned embeddings.

**4D Positional Encoding and Spatial Augmentation.**   We start with the spatial positions of the EEG electrodes $\mathbf{P} \in \mathbb{R}^{C \times 3}$, where each row of $\mathbf{P}$ contains the $(x, y, z)$ coordinates of a channel, to which we add Gaussian noise with standard deviation $\sigma_{\mathrm{noise}}$. This improves generalization to diverse electrode positions and ensures robustness to variability in head size or electrode placement. We extend $\mathbf{P}$ with a temporal component, resulting in $\mathbf{P}_{\mathrm{ext}} \in \mathbb{R}^{C \times p \times 4}$, where $p$ is the number of

patches obtained from segmenting EEG signal, as defined in Section 2.1. The temporal dimension is represented as discrete values from 1 to $p$, scaled by a factor $s_t$ to ensure a scale similar to the spatial dimensions.

**4D Fourier-Based Position Encoding.** Building on the 2D approach proposed by Défossez et al. (2023), we extend the Fourier positional encoding method to 4D in our encoding strategy, as follows. Each positional component $(x, y, z, t)$ of $\mathbf{P}_{\text{ext}}$ is projected into a multi-frequency space, using $n_{\text{freq}}$ frequencies per dimension. The frequency assignment follows a Cartesian product structure, *i.e.*, all combinations of frequencies across the four dimensions contribute to the encoding, resulting in a flattened vector of dimension $n_{\text{freq}}^4$. A hierarchical periodicity emerges: the period of $x$ is $n_{\text{freq}}^1$, of $y$ is $n_{\text{freq}}^2$, of $z$ is $n_{\text{freq}}^3$, and of $t$ is $n_{\text{freq}}^4$. Then, applying sine and cosine transformations doubles the embedding size, producing a positional vector of dimension $2 \cdot n_{\text{freq}}^4$. We ensure that the embedding dimension matches the hidden size required by the 4DPE module, with $n_{\text{freq}} \in \{3, 4, 5\}$ resulting in the final embedding $\mathbf{F}_{\text{pe}} \in \mathbb{R}^{C \times p \times D_E}$. The 4D encoding adds minimal compute overhead, with sinusoidal computations and a small linear layer. Computational cost scales linearly with the number of input tokens (channels $\times$ temporal patches) and is negligible relative to the transformer backbone.

**Final Adjusted Position Encoding.** To complement the fixed Fourier features, we also process $\mathbf{P}_{\text{ext}}$ through a linear layer followed by GELU (Hendrycks and Gimpel, 2016) and LayerNorm (Lei Ba et al., 2016), producing a learnable representation $\mathbf{F}_{\text{lin}} \in \mathbb{R}^{C \times p \times D_E}$. This component adapts the positional encoding to the specific dataset and task, and can compensate for any truncation in the Fourier basis. The final positional encoding is given by $\mathbf{P}_{\text{enc}} = \text{LayerNorm}(\mathbf{F}_{\text{pe}} + \mathbf{F}_{\text{lin}})$, combining the structured inductive bias of Fourier features with the flexibility of learned adaptation. This vector is added to the non-masked patch embeddings before being passed to the transformer encoder similarly to MAE (He et al., 2022), and is consistent with standard absolute positional encoding practices (Vaswani, 2017). The ablation study in Table 19 confirms that this method outperforms both fixed learnable and purely MLP-based positional encoding schemes.

## 2.3 Transformer

Our model extends the standard Transformer architecture (Vaswani, 2017) with enhancements that improve efficiency and stability. We use RMSNorm (Zhang and Sennrich, 2019) in lieu of LayerNorm as a **normalization layer** for better training stability, and choose GEGLU (Shazeer, 2020) as the **activation** function in the feed-forward network (FFN) layers as it outperforms standard GELU through more expressive gating mechanisms (Geiping and Goldstein, 2023). This choice is further supported by the ablation results in Table 20. Our **FFN layers** follow a two-layer structure with an expansion ratio of $\frac{8}{3}$, consistent with designs from LLaMA (Touvron et al., 2023), Qwen (Bai et al., 2023) or Mistral (Jiang et al., 2023). Following Dayma et al. (2021), we **remove bias terms** from all linear layers except the final decoder layer. This reallocates the parameter budget to linear transformations, improving efficiency. We use **Flash Attention v2** (Dao, 2024) for memory and computational efficiency in the attention as it reduces the softmax overhead and ensures scalability to long sequences, while maintaining the core transformer formulation.

## 2.4 Masked EEG Reconstruction Methodology

During pretraining, our model reconstructs EEG signal of masked patches using information from the visible, unmasked patches. The overall pretraining framework is illustrated in Figure 1.

Let $\mathbf{P}_{\text{m}} \in \mathbb{R}^{N_{\text{m}} \times w}$, and $\mathbf{P}_{\overline{\text{m}}} \in \mathbb{R}^{N_{\overline{\text{m}}} \times w}$ denote the masked and visible patches, respectively, with $N_{\text{m}}$ and $N_{\overline{\text{m}}}$ as defined in Section 2.1. The associated patch embeddings are denoted as $\mathbf{E}_{\text{m}}$ for masked patches and $\mathbf{E}_{\overline{\text{m}}}$ for visible patches.

We adopt the MAE structure from He et al. (2022), with a larger encoder and a lighter decoder each following the architecture described in Section 2.3. Only the embeddings of the visible patches $\mathbf{E}_{\overline{\text{m}}}$, enriched with their positional encodings are passed through the encoder, to produce latent representations $\mathbf{F}_{\overline{\text{m}}}$. Masked patches are represented using a learned embedding, repeated $N_{\text{m}}$ times and also augmented with positional encodings. Before entering the decoder, positional encodings are re-added to both visible and masked latent patches. Together, they form the decoder input from which the raw EEG signal of the masked patches is reconstructed.

Unlike the original MAE, which uses a separate set of fixed positional encodings for the decoder, we reuse the same encoding for both the encoder and decoder. This design ensures flexibility for processing EEG signals with varying temporal lengths and electrode configurations.

The output of the decoder transformer, is passed through a linear projection layer that maps latent patches back into the signal space, reconstructing the raw EEG signal of the masked patches. Reconstructed patches minimize the $L_1$ loss relative to the original raw EEG patches:

$$\mathcal{L} = \frac{1}{|\mathbf{P}_m|} \sum_{i \in \mathbf{P}_m} \left\| \hat{\mathbf{P}}_m^{(i)} - \mathbf{P}_m^{(i)} \right\|_1 \tag{1}$$

where $\hat{\mathbf{P}}_m^{(i)}$ represents the reconstructed signal for patch $i$, and $\mathbf{P}_m^{(i)}$ is the original signal. We chose $L_1$ loss over $L_2$ due to the inherently noisy nature of EEG signals. While $L_2$ amplifies the influence of noise, $L_1$ loss offers greater robustness by reducing the impact of outliers.

In addition to the main reconstruction loss, we introduce a secondary task that reconstructs masked patches from a compact global representation. We apply attention pooling over the outputs of all Multi-Head Attention (MHA) layers in the encoder: the output tokens (after FFN) from each MHA block are concatenated and attended by a learned query token. This pooled token is then repeated, enriched with positional encodings, and passed through a 2-layer FFN to reconstruct the masked patches. As with the primary loss, we use $L_1$ loss for reconstruction. The total loss is a weighted sum: Loss = Primary Loss + $\lambda \cdot$ Secondary Loss

This secondary loss encourages the encoder to distribute useful information across all layers, mitigating over-specialization in the final layer and yielding more generalizable representations.

The secondary loss mitigates a limitation of the MAE framework: the final encoder layer can overfit to the reconstruction task, especially with a shallow decoder (He et al., 2022). By pooling features across all transformer layers (Alkin et al., 2024), the learned token captures a compact, global EEG representation, encouraging more balanced use of the encoder depth. This leads to stronger, more generalizable features for downstream tasks like linear probing, few-shot learning, and transfer without fine-tuning.

After the pretraining phase, the decoder is discarded, and only the encoder is used. In this case, no embeddings are masked, *i.e.*, $\mathbf{P}_m = \mathbf{E}_m = \emptyset$. All patches are processed as usual by retaining their associated positional encoding.

To avoid confusion regarding terminology, we clarify that the terms "encoder" and "decoder" are used here in the context of masked auto-encoders (MAE), not in the autoregressive Transformer sense. All Transformer blocks in REVE are non-causal and operate within a standard encoder-style attention pattern; no autoregressive training is involved. The "decoder" refers solely to the lightweight reconstruction head used to recover masked EEG segments during self-supervised pretraining.

## 3 Experiments

### 3.1 Pretraining

This section outlines the data sources and preprocessing steps used for pretraining, followed by our strategy for scalable and effective representation learning across diverse datasets.

#### 3.1.1 Dataset Collection & Preprocessing

To enable large-scale pretraining, we assembled a massive and diverse collection of EEG recordings from open-source or request-accessible datasets. It comprises 19 TB of raw data, spanning 24,274 subjects, 150,833 unique sessions, and 61,415 hours of recordings drawn from 92 different sources, including OpenNeuro (Markiewicz et al., 2021), MOABB (Aristimunha et al., 2023), and TUH (Obeid and Picone, 2016). To our knowledge, this is the largest and most diverse EEG dataset assembled for training a foundation model. The most extensive prior effort, by Yuan et al. (2024a), comprised approximately 40,000 hours of recordings, but primarily relied on intracranial EEG (iEEG) rather than non-invasive EEG. A summary of the dataset composition and a full list of included sources are provided in Appendix B. While the majority of the data consists of clinical EEG recordings, we also include a substantial subset of cognitive and BCI-related data which, although smaller in proportion,

tend to be cleaner and more diverse. We also collected electrode positional information for each recording. When 3D coordinates were available, they were used directly; otherwise, positions were inferred from standard labels. Channels without identifiable names or positional data were excluded. The dataset spans a wide range of EEG systems and formats—including BrainVision, BioSemi, EDF, GDF, and EEGLAB, with most recordings adhering to the 10-5 system (Oostenveld and Praamstra, 2001). In total, the dataset includes 396 unique electrode names.

Our preprocessing pipeline is designed to preserve signal diversity and prioritize robustness when scaling. We only removed recordings shorter than 10 seconds, and those used in downstream tasks. Remaining signals were resampled to 200 Hz, band-pass filtered (0.5–99.5 Hz), and converted to float32, resulting in a 6 TB dataset. To address amplitude variations across recordings, we applied Z-score normalization with statistics computed across the recording sessions to ensure robust statistics. After normalization, values exceeding 15 standard deviations were clipped, as in Défossez et al. (2023). Unlike CBraMod (Wang et al., 2024b), which excluded signals above 100 $\mu$V, our approach retains them, resulting in about 60,000 hours of EEG, compared to 9,000 in CBraMod and 2,534 in LaBraM.

### 3.1.2 Pretraining Strategy & Scaling

### 3.2 Training and Scaling Strategy

We present the training procedure used for pretraining the Small model and detail how it scales to larger architectures under constrained resources. Our training framework builds upon recent advances in state-of-the-art NLP methodologies (Warner et al., 2024). We use the StableAdamW (Wortsman et al., 2023) optimizer, designed for low precision frameworks and improved stability, thanks to the Adafactor-style gradient clipping (Shazeer and Stern, 2018). Table 5 of the Appendix lists the optimizer hyperparameters.
The learning rate follows a Warmup Stable Decay (trapezoidal) schedule (Hu et al., 2024), known for its robustness to learning rate variations (Hägele et al., 2024). We use a linear warmup over 10% of the first epoch, followed by 80% at peak LR, and a linear decay to 1% of the maximum. Unlike one-cycle schedules that reset every epoch, our cyclic trapezoidal variant allows multiple cooldown phases across epochs, particularly beneficial for EEG training where masked token sampling introduces variability. We apply Megatron-style initialization (Shoeybi et al., 2019) with a standard deviation of 0.02 for all transformer layers and the mask token, ensuring stable dynamics. Other parameters use PyTorch's default initializations.

A key factor for the success of foundation models is the simultaneous scaling of both training datasets and model architectures (Touvron et al., 2023). We describe our scaling methodology to maximize computational efficiency and accommodate larger models within constrained resources. To scale model capacity, we adjust depth, width, and number of attention heads while maintaining a fixed FFN ratio. Table 6 of the Appendix summarizes these configurations. This scaling strategy enables efficient capacity expansion while preserving architectural consistency across model sizes.
Recent advances in NLP provide strong theoretical and empirical evidence for the existence of scaling laws (Kaplan et al., 2020; Hoffmann et al., 2022), which govern the relationship between model size, training dynamics, optimization and initialization hyperparameters. We follow the power law $\eta \propto D^{\alpha_D}$, with $\alpha_D = -0.90$ and $D$ the model dimension, for the learning rate, as derived in Everett et al. (2024). The optimal LR is first swept on the small model and then scaled accordingly.
To efficiently train models, we use data parallelism, maintaining a constant batch size by reducing per-GPU loads for large models. A load-aware data-shuffling strategy groups samples by electrode count, shuffles within and across buckets, and balances batches across GPUs to avoid bottlenecks, for constant optimization steps and maximized throughput.
Although scaling laws exist for adjusting AdamW momentum terms (Malladi et al., 2022), our use of a constant effective batch size across models allows us to fix $\beta_1$ and $\beta_2$. Regarding initialization, while Hägele et al. (2024) suggests scaling $\sigma_{\text{init}} \propto D^{-0.5}$, our width increase (from 200 to 1,216) leads us to keep $\sigma_{\text{init}} = 0.02$ fixed across scales.

### 3.3 Downstream tasks

**Downstream task datasets**    To evaluate the performance and generalizability of our EEG foundation model, we perform extensive assessments across 10 diverse downstream tasks, selected to ensure

comparability with existing models in the field. These tasks span a variety of EEG-based applications, including sleep staging, emotion and event classification, detection of stress and mental disorder , across the following datasets: PhysioNet-MI (Goldberger et al., 2000), BCIC-IV-2a (Tangermann et al., 2012), TUEV (Obeid and Picone, 2016), TUAB (Obeid and Picone, 2016), HMC (Alvarez-Estevez and Rijsman, 2021), ISRUC (Khalighi et al., 2016), FACED (Chen et al., 2023), Mumtaz (Mumtaz, 2016), Mental Arithmetic (MAT) (Zyma et al., 2019), and BCI2020-IV-3 (Jeong et al., 2022). A summary of these datasets is provided in Table 1, with a more detailed description available in the Appendix.

Table 1: Overview of downstream tasks and datasets.

| Task | Dataset | # Channels | Duration | # Samples | Rate | # Classes |
|---|---|---|---|---|---|---|
| Motor Imagery | PhysioNet-MI | 64 | 4s | 9,837 | 160Hz | 4 |
| | BCIC-IV-2a | 22 | 4s | 5,184 | 250Hz | 4 |
| Event Type | TUEV | 16 | 5s | 112,491 | 256Hz | 6 |
| Abnormal detection | TUAB | 16 | 10s | 409,455 | 256Hz | 2 |
| Sleep staging | HMC | 4 | 30s | 137,243 | 256 Hz | 5 |
| | ISRUC | 6 | 30s | 89,240 | 200Hz | 5 |
| Emotion recognition | FACED | 32 | 10s | 10,332 | 250Hz | 9 |
| Mental disorder | Mumtaz | 19 | 5s | 7,143 | 256Hz | 2 |
| Mental stress | MAT | 20 | 5s | 1,707 | 500Hz | 2 |
| Imagined speech | BCIC2020-3 | 64 | 3s | 6,000 | 256Hz | 5 |

Our evaluation process maintains strict consistency with prior works by adhering to the same train/val/test splits used in earlier studies, ensuring that our results are directly comparable to baseline models. Specifically, we follow the protocols from CBraMod (Wang et al., 2024b), LaBraM (Jiang et al., 2024), and BIOT (Yang et al., 2024), guaranteeing fair comparisons across tasks. For fairness in preprocessing, we adopt the same pipeline as the baselines. A notable correction was made for the ISRUC dataset, where we identified and removed a bug in the baseline code involving the inclusion of a chin electrode instead of an EEG electrode. Our results for REVE exclude the chin electrode, aligning with proper electrode placement.

**Finetuning** Fine-tuning EEG-based models presents unique challenges due to the small size of available datasets and the high noise levels in EEG recordings. Unlike large-scale vision datasets, EEG datasets are often limited in size, subject-dependent, and prone to distribution shifts across different recording setups. Effective fine-tuning must therefore maximize generalization while mitigating the risk of overfitting. To address this, we adopt a two-step fine-tuning strategy, incorporating techniques specifically designed to enhance stability and adaptability. This includes the use of parameter-efficient fine-tuning techniques (Suzumura et al., 2024) tailored to this domain.

For downstream classification tasks, the two-step strategy, inspired by Kumar et al. (2022), goes as follow: We first train a linear probe while keeping the encoder frozen, aligning the classifier with the pretrained feature space. Next, we unfreeze the encoder and fine-tune the entire network for task-specific adaptation, preserving the robustness of the pretrained model. Importantly, this two-step strategy is implemented as a single continuous training run, where the backbone is initially frozen (i.e., only the head is trained) and later unfrozen. This approach is well-suited for EEG data, where distributions can shift significantly across datasets. We employ dropout and Mixup (Zhang et al., 2018) as data augmentation for improved robustness. To further mitigate catastrophic forgetting and improve efficiency, we integrate Low-Rank Adaptation (LoRA) into the attention blocks, within the query, key, value, and output (QKVO) projection layers. Instead of fine-tuning the entire model, LoRA introduces trainable low-rank matrices that enable effective adaptation while preserving the integrity of the pretrained model's knowledge (Hu et al., 2022).

Each training step includes a warmup phase (Kalra and Barkeshli, 2024) followed by a cooldown phase. The cooldown phase employs a Reduce-on-Plateau learning rate scheduler, which dynamically lowers the learning rate when training convergence slows to preventing overfitting.

To further enhance robustness, we explore model souping (Wortsman et al., 2022), which averages the weights of multiple fine-tuning runs to improve accuracy. Given the stochasticity and noise inherent in EEG datasets, souping smooths gradients and reduces variance across different fine-tuning

trajectories. Our experiments confirm that this approach enhances generalization and produces more stable performance across diverse EEG tasks.

By integrating structured fine-tuning with data augmentation, LoRA and model souping, our approach effectively addresses the small-scale and noisy nature of EEG datasets. These techniques effectively ensure robust and generalized adaptation to downstream tasks.

# 4 Results and Discussion

We evaluate REVE against non-foundation and foundation model baselines on the previously discussed datasets.

**Non-Foundation Models:** We compare to EEGNet (Lawhern et al., 2018), EEGConformer (Song et al., 2022), SPaRCNet (Jing et al., 2023), ContraWR (Yang et al., 2021), CNN-Transformer (Peh et al., 2022), FFCL (Li et al., 2022), and ST-Transformer (Song et al., 2021).

**Foundation Models:** We compare to BIOT (Yang et al., 2024), LaBraM (Jiang et al., 2024) and CBraMod (Wang et al., 2024b). We report results displayed in existing studies.

We report the balanced accuracy for each dataset and provide additional evaluation metrics in the appendix.

Table 2: Balanced accuracy ($\pm$ std) of different methods across 9 EEG classification task

| Methods | TUAB | TUEV | PhysioNetMI | BCI-IV-2a | FACED |
|---|---|---|---|---|---|
| EEGNet | $0.7642 \pm 0.0036$ | $0.3876 \pm 0.0143$ | $0.5814 \pm 0.0125$ | $0.4482 \pm 0.0094$ | $0.4090 \pm 0.0122$ |
| EEGConformer | $0.7758 \pm 0.0049$ | $0.4074 \pm 0.0164$ | $0.6049 \pm 0.0104$ | $0.4696 \pm 0.0106$ | $0.4559 \pm 0.0125$ |
| SPaRCNet | $0.7896 \pm 0.0018$ | $0.4161 \pm 0.0262$ | $0.5932 \pm 0.0152$ | $0.4635 \pm 0.0117$ | $0.4673 \pm 0.0155$ |
| ContraWR | $0.7746 \pm 0.0041$ | $0.4384 \pm 0.0349$ | $0.5892 \pm 0.0133$ | $0.4678 \pm 0.0125$ | $0.4887 \pm 0.0078$ |
| CNN-Transformer | $0.7777 \pm 0.0022$ | $0.4087 \pm 0.0161$ | $0.6053 \pm 0.0118$ | $0.4600 \pm 0.0108$ | $0.4697 \pm 0.0132$ |
| FFCL | $0.7848 \pm 0.0038$ | $0.3979 \pm 0.0104$ | $0.5726 \pm 0.0092$ | $0.4470 \pm 0.0143$ | $0.4673 \pm 0.0158$ |
| ST-Transformer | $0.7966 \pm 0.0023$ | $0.3984 \pm 0.0228$ | $0.6035 \pm 0.0081$ | $0.4575 \pm 0.0145$ | $0.4810 \pm 0.0079$ |
| BIOT | $0.7959 \pm 0.0057$ | $0.5281 \pm 0.0225$ | $0.6153 \pm 0.0154$ | $0.4748 \pm 0.0093$ | $0.5118 \pm 0.0118$ |
| LaBraM-Base | $0.8140 \pm 0.0019$ | $0.6409 \pm 0.0065$ | $0.6173 \pm 0.0122$ | $0.4869 \pm 0.0085$ | $0.5273 \pm 0.0107$ |
| CbraMod | $0.8289 \pm 0.0022$ | $0.6671 \pm 0.0107$ | $0.6417 \pm 0.0091$ | $0.5138 \pm 0.0066$ | $0.5509 \pm 0.0089$ |
| REVE-Base | $\mathbf{0.8315} \pm 0.0014$ | $\mathbf{0.6759} \pm 0.0229$ | $\mathbf{0.6480} \pm 0.0140$ | $\mathbf{0.6396} \pm 0.0095$ | $\mathbf{0.5646} \pm 0.0164$ |
| | ISRUC | Mumtaz | MAT | BCI-2020-3 | **Average** |
| EEGNet | $0.7154 \pm 0.0121$ | $0.9232 \pm 0.0104$ | $0.6770 \pm 0.0116$ | $0.4413 \pm 0.0096$ | $0.5941 \pm 0.0037$ |
| EEGConformer | $0.7400 \pm 0.0133$ | $0.9308 \pm 0.0117$ | $0.6805 \pm 0.0123$ | $0.4506 \pm 0.0133$ | $0.6128 \pm 0.0044$ |
| SPaRCNet | $0.7487 \pm 0.0075$ | $0.9316 \pm 0.0095$ | $0.6879 \pm 0.0107$ | $0.4426 \pm 0.0156$ | $0.6156 \pm 0.0047$ |
| ContraWR | $0.7402 \pm 0.0126$ | $0.9195 \pm 0.0115$ | $0.6631 \pm 0.0097$ | $0.4257 \pm 0.0162$ | $0.6119 \pm 0.0053$ |
| CNN-Transformer | $0.7363 \pm 0.0087$ | $0.9305 \pm 0.0068$ | $0.6779 \pm 0.0268$ | $0.4533 \pm 0.0092$ | $0.6133 \pm 0.0045$ |
| FFCL | $0.7277 \pm 0.0182$ | $0.9314 \pm 0.0038$ | $0.6798 \pm 0.0142$ | $0.4678 \pm 0.0197$ | $0.6085 \pm 0.0044$ |
| ST-Transformer | $0.7381 \pm 0.0205$ | $0.9135 \pm 0.0103$ | $0.6631 \pm 0.0173$ | $0.4126 \pm 0.0122$ | $0.6071 \pm 0.0048$ |
| BIOT | $0.7527 \pm 0.0121$ | $0.9358 \pm 0.0052$ | $0.6875 \pm 0.0186$ | $0.4920 \pm 0.0086$ | $0.6438 \pm 0.0044$ |
| LaBraM-Base | $0.7633 \pm 0.0102$ | $0.9409 \pm 0.0079$ | $0.6909 \pm 0.0125$ | $0.5060 \pm 0.0155$ | $0.6653 \pm 0.0031$ |
| CBraMod | $\mathbf{0.7865} \pm 0.0110$ | $0.9560 \pm 0.0056$ | $0.7256 \pm 0.0132$ | $0.5373 \pm 0.0108$ | $0.6898 \pm 0.0031$ |
| REVE-Base | $0.7819 \pm 0.0078^3$ | $\mathbf{0.9644} \pm 0.0097$ | $\mathbf{0.7660} \pm 0.0355$ | $\mathbf{0.5635} \pm 0.0123$ | $\mathbf{0.7150} \pm 0.0057$ |

Table 2 shows that REVE achieves state-of-the-art performance on the downstream tasks in this study, with an average gain of 2.5%, compared to CBraMod the highest performing baseline. The results on ISRUC and HMC (Appendix C.6) show that the model effectively generalizes beyond the 10-second segments it was pretrained on, performing well on tasks with 30-second inputs, which highlights the strength of our positional encoding method. The results on TUEV highlight the model's ability to generalize to unseen electrode configurations, including bipolar setups never encountered during training.

In addition to the detailed evaluation metrics provided in Appendix C, we report the performance of the Large model across our downstream tasks in Table 4. We observe that the Large model consistently produces richer embeddings, leading to improved linear probing performance compared to the Base model. Model souping consistently improved performance, averaging a 1.5% gain when

---

[3]NB: our preprocessing pipeline is different from the baseline and fixes a potential bug

Table 3: Impact of pretraining (PT) and weight freezing on REVE and baselines for PhysioNet-MI

| Settings | PhysioNet-MI, 4-class | | |
|---|---|---|---|
| | **Balanced Accuracy** | **Cohen's Kappa** | **Weighted F1** |
| CBraMod (w/ PT) | $0.6417 \pm 0.0091$ | $0.5222 \pm 0.0169$ | $0.6427 \pm 0.0100$ |
| BIOT (w/ PT) | $0.6153 \pm 0.0154$ | $0.4875 \pm 0.0272$ | $0.6158 \pm 0.0197$ |
| LaBraM-Base (w/ PT) | $0.6173 \pm 0.0122$ | $0.4912 \pm 0.0192$ | $0.6177 \pm 0.0141$ |
| REVE-Base (w/ PT) | **0.6480** $\pm 0.0140$ | **0.5306** $\pm 0.0187$ | **0.6484** $\pm 0.0170$ |
| CBraMod (w/o PT) | $\underline{0.6196} \pm 0.0143$ | $\underline{0.4994} \pm 0.0289$ | $\underline{0.6289} \pm 0.0179$ |
| REVE-Base (w/o PT) | $0.5409 \pm 0.0094$ | $0.3879 \pm 0.0125$ | $0.5421 \pm 0.0101$ |
| Cbramod (Frozen) | $0.3845 \pm 0.0345$ | $0.2983 \pm 0.0498$ | $0.3946 \pm 0.0378$ |
| BIOT (Frozen) | $0.3698 \pm 0.0318$ | $0.2703 \pm 0.0472$ | $0.3723 \pm 0.0364$ |
| LaBraM (Frozen) | $0.3715 \pm 0.0458$ | $0.2814 \pm 0.0586$ | $0.3796 \pm 0.0472$ |
| REVE-Base (Frozen) | $\underline{0.5371} \pm 0.0052$ | $\underline{0.3827} \pm 0.0070$ | $\underline{0.5376} \pm 0.0033$ |

Table 4: **Linear probing** results on downstream tasks for REVE and CBraMod models with (Pool) and without pooling across multiple EEG downstream tasks. Best results are highlighted in bold. To ensure a fair comparison, we reproduced CBraMod (Wang et al., 2024b) using their official code and pretrained checkpoint, carefully following their classification pipeline (notably, no pooling) and matched architectural details to avoid any bias.

| Dataset | REVE-B (Pool) | REVE-B | REVE-L (Pool) | REVE-L | CBraMod (Pool) | CBraMod |
|---|---|---|---|---|---|---|
| Mumtaz | $0.962 \pm 0.003$ | $0.931 \pm 0.021$ | **0.985 ± 0.006** | $0.980 \pm 0.009$ | $0.859 \pm 0.009$ | $0.907 \pm 0.027$ |
| M. Arithmetic | $0.725 \pm 0.010$ | **0.740 ± 0.073** | $0.712 \pm 0.008$ | $0.665 \pm 0.103$ | $0.500 \pm 0.000$ | $0.605 \pm 0.020$ |
| TUAB | $0.810 \pm 0.007$ | $0.809 \pm 0.004$ | **0.821 ± 0.004** | $0.809 \pm 0.004$ | $0.500 \pm 0.000$ | $0.500 \pm 0.000$ |
| PhysioNetMI | $0.537 \pm 0.005$ | $0.510 \pm 0.012$ | $0.551 \pm 0.001$ | **0.617 ± 0.000** | $0.256 \pm 0.002$ | $0.531 \pm 0.015$ |
| BCIC-IV-2a | $0.432 \pm 0.004$ | $0.517 \pm 0.015$ | $0.534 \pm 0.001$ | **0.603 ± 0.011** | $0.287 \pm 0.023$ | $0.376 \pm 0.006$ |
| ISRUC | $0.697 \pm 0.011$ | $0.662 \pm 0.030$ | $0.743 \pm 0.004$ | **0.758 ± 0.001** | $0.407 \pm 0.049$ | $0.430 \pm 0.043$ |
| HMC | $0.647 \pm 0.008$ | $0.604 \pm 0.008$ | $0.703 \pm 0.003$ | **0.710 ± 0.007** | $0.368 \pm 0.001$ | $0.538 \pm 0.009$ |
| BCIC2020-3 | $0.234 \pm 0.009$ | **0.390 ± 0.017** | $0.274 \pm 0.001$ | $0.378 \pm 0.021$ | $0.214 \pm 0.003$ | $0.374 \pm 0.007$ |
| TUEV | $0.592 \pm 0.008$ | $0.508 \pm 0.073$ | **0.630 ± 0.003** | $0.550 \pm 0.014$ | $0.219 \pm 0.009$ | $0.482 \pm 0.037$ |
| Faced | $0.240 \pm 0.010$ | $0.422 \pm 0.028$ | $0.283 \pm 0.003$ | **0.469 ± 0.007** | $0.117 \pm 0.005$ | $0.261 \pm 0.013$ |
| **Avg.** | 0.586 | 0.609 | 0.623 | **0.654** | 0.373 | 0.501 |

combining at least 5 Base or Large models. For example, REVE-Base achieved 69.6% balanced accuracy on TUEV using the 10 models from Table 2. However, souping showed limited benefits for the small models and sometimes led to negative outcomes.

Table 3 highlights the importance of REVE's pretraining phase. Without pretraining, CBraMod outperforms REVE by at least 8%. However, pretraining improves REVE-Base by 11%, while CBraMod gains only 2%, a trend also observed in the LaBraM paper. This suggests that REVE benefits more significantly from pretraining, whereas other models derive most of their performance from architectural design rather than pretraining learned representations. A key advantage of REVE is its ability to produce high-quality latent spaces without heavy fine-tuning, as evidenced by linear probing results in Table 4: REVE consistently outperforms CBraMod across all downstream tasks and model sizes, with REVE-Large achieving nearly 17% higher performance. These results also highlight REVE's ability to scale effectively with model size, yielding richer and more generalizable embeddings as capacity increases. Providing rich, ready-to-use embeddings is crucial for enabling zero-shot analysis, faster BCI calibration, and improved performance in low-data or sparsely annotated settings. REVE also benefits from its spatial encoding strategy, which enables transfer across diverse EEG configurations. In Appendix D, we further demonstrate the contribution of our secondary loss function, a novel component of our framework, which proves particularly effective in frozen-feature scenarios. The secondary objective reconstructs masked tokens using a compressed, global representation from attention pooling. This pooling acts as an information bottleneck, forcing the model to distill key information from the entire input sequence into a single vector. As shown by Table 17, the secondary loss mainly improves the quality of the frozen embeddings of the model.

## 5 Limitations and Future Work

The model has some limitations, requiring signals to be at least one second and multiples of one second. A way to address this could be to leverage padding with causal masking.

While the focus has been on collecting large EEG datasets for pretraining, an important next step could be to curate this data more selectively. This includes removing low-quality recordings, balancing distributions, and identifying representative subsets, especially given the inherently noisy nature of EEG signals. Our current pretraining corpus aggregates 92 publicly available EEG datasets spanning over 25,000 subjects, which helps reduce overfitting to any single source. However, most public EEG data originates from North America and Europe, resulting in limited demographic diversity—a key limitation that calls for broader, more equitable data collection efforts. To partially mitigate such imbalances, we leverage self-supervised learning (MAE), which has been shown to be robust to long-tailed and heterogeneous data distributions (Xu et al., 2023). Targeted selection strategies, combined with robust SSL objectives, could help focus on the most informative and complementary data for building stronger, fairer, and more efficient foundation models. Thanks to its flexibility in handling any EEG configuration, REVE could itself guide this curation process.

We also plan to extend our study to diverse tasks, including zero-/few-shot regimes. This first iteration uses a simple MAE approach and a standard transformer, but future improvements could leverage more advanced SSL techniques and architectures. We release the model's code, weights and guidelines for adapting it to mainstream EEG tasks. In parallel, our findings point toward the presence of scaling effects in EEG foundation models. Identifying precise scaling laws that capture how model size, data volume, and downstream performance interact would be valuable for future work.

## 6 Conclusion

EEG research has lacked a foundation model that transfers robustly across devices, montages, and tasks—especially under linear probing. REVE contributes to bridging this gap. Trained on 60,000 hours from 92 datasets and 25,000 subjects, REVE combines a 4D Fourier positional encoding that natively supports arbitrary electrode layouts and sequence lengths with masked autoencoding enhanced by spatio-temporal block masking and a global-token secondary loss. Across 10 benchmarks, it sets a new state of the art (average +2.5% balanced accuracy over prior foundation models), delivers up to 17% gains in linear probing, and generalizes to unseen/bipolar montages and longer inputs than used in pretraining. These properties enable faster BCI calibration, more reliable cross-site clinical deployment, and standardized embeddings for downstream analytics. We release code, weights, loaders for arbitrary 3D coordinates, and training/eval recipes. We invite the community to extend REVE to broader populations and modalities (MEG/iEEG/OPM-MEG), and to co-build a cross-montage benchmark for fair, scalable EEG evaluation.

## 7 Acknowledgments

This research was supported by the French National Research Agency (ANR) through its AI@IMT program and grant ANR-24-CE23-7365, as well as by a grant from the Brittany region. Further support was provided by a Discovery Grant from the Natural Sciences and Engineering Research Council of Canada (NSERC), by funding from the Canada Research Chairs program and the Fonds de recherche du Québec – Nature et technologies (FRQ-NT). This work was granted access to the HPC resources of IDRIS under the allocation 2024-AD011015237R1 made by GENCI, as well as HPC provided by Digital Alliance Canada.

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

# Appendix

## A    Configurations

We report the hyperparameters used to train the REVE suite of models, including data preprocessing steps, self-supervised masking configurations, and optimizer settings governing the training dynamics. Notations are consistent with those in the main text.

Table 5: Exhaustive list of all hyperparameter values

| Variable | Meaning | Value |
|---|---|---|
| **Data preprocessing** | | |
| $w$ | Window size | 1s |
| $o$ | Overlap | 0.1s |
| $\sigma_{\text{noise}}$ | Position noise std | 0.25cm |
| **Masking parameters** | | |
| $M_r$ | Total masking ratio | 55% |
| $R_s$ | Spatial masking radius | 3 cm |
| $R_t$ | Temporal masking radius | 3 seconds |
| $D_r$ | Dropout ratio | 10% |
| $R_d$ | Dropout spatial radius | 4 cm |
| **Training dynamics** | | |
| | Optimizer | StableAdamW |
| | Scheduler | Warmup Stable Decay |
| $\eta$ | Peak learning rate | $\eta = 2.4 \cdot 10^{-4}$ |
| $\beta_1, \beta_2$ | Momentum constants | 0.9, 0.95 |
| $\varepsilon$ | Numerical stability bias | $10^{-9}$ |
| $\sigma_{\text{init}}$ | Initialization std | 0.02 |
| | Batch size | 4,096 |
| $\lambda$ | Secondary loss multiplier | 0.1 |

We report how the scaled number of parameters is allocated across our models. We also indicate the number of Fourier frequencies encoded (see Section 2.2). Note that no frequency truncation was required, as we closely matched the hidden dimension of our models to the number of components generated by the 4D PE module.

Table 6: Summary of encoder configurations for different sizes

| Size | depth | n_heads | dim | params (M) | $n_{\text{freq}}$ |
|---|---|---|---|---|---|
| Small | 4 | 8 | 512 | 12 | 4 |
| Base | 22 | 8 | 512 | 69 | 4 |
| Large | 22 | 19 | 1250 | 408 | 5 |

## B    Pretraining dataset

We include a summarized description of the pretraining dataset composition, grouped by category, platform of origin and number of channels. The final dataset spans 61,415 hours of recordings from 92 datasets encompassing 24,274 subjects.

Table 7: Detailed overview of the pretraining datasets.

| Group | Subjects | Duration (hours) | Datasets |
|---|---|---|---|
| **Category** | | | |
| BCI | 791 | 457 | 28 |
| Cognition | 4,193 | 10,376 | 56 |
| Clinic | 19,290 | 50,581 | 8 |
| **Platform** | | | |
| TUH | 14,987 | 26,847 | 1 |
| Physionet | 607 | 22,707 | 2 |
| OpenNeuro | 4153 | 10,194 | 56 |
| MOABB | 711 | 384 | 27 |
| Other | 3,802 | 1,250 | 6 |
| **Channels** | | | |
| $[3-30[$ | 19,871 | 50,870 | 31 |
| $[30-80[$ | 1,781 | 1,516 | 48 |
| $[80-129]$ | 2,622 | 9,027 | 13 |
| **Total** | **24,274** | **61,415** | **92** |

We provide and exhaustive list of the datasets in the pretraining set, along with their respective licenses.

**MOABB (Aristimunha et al., 2023):** AlexMI (Barachant, 2012), BNCI2014004 (Leeb et al., 2007), BNCI2015001 (Faller et al., 2012), BNCI2015004 (Scherer et al., 2015), Cho2017 (Cho et al., 2017), Lee2019MI (Lee et al., 2019), Liu2024 (Liu et al., 2024), Ofner2017 (Ofner et al., 2017), Shin2017A (Shin et al., 2016), Weibo2014 (Yi et al., 2014), Zhou2016 (Zhou et al., 2016), Schirrmeister2017 (Schirrmeister et al., 2017), Kalunga2016 (Kalunga et al., 2015), Lee2019SSVEP (Lee et al., 2019), Nakanishi2015 (Nakanishi et al., 2015), BI2014a (Korczowski et al., 2019b), BI2014b (Korczowski et al., 2019c), BNCI2014008 (Riccio et al., 2013), BNCI2014009 (Aricò et al., 2014), BNCI2015003 (Guger et al., 2009), EPFLP300 (Hoffmann et al., 2008), BI2015a (Korczowski et al., 2019a), BI2015b (Korczowski et al., 2019c), Sosulski2019 (Sosulski et al., 2021), Lee2019ERP (Lee et al., 2019)
MOABB is under a BSD 3-Clause License.

**Physionet (Goldberger et al., 2000):** Siena (Detti, 2020; Detti et al., 2020), under the Creative Commons Attribution 4.0 International Public License, ICARE (Amorim et al., 2023) under the Creative Commons Attribution-NonCommercial-ShareAlike 4.0 International Public License,

**OpenNeuro:** ds004706 (Rudoler et al., 2023), ds004582 (Makowski et al., 2023), ds004356 (Shan et al., 2022), ds004817 (Grootswagers et al., 2023b), ds005189 (Helbing et al., 2024), ds003887 (Shatek et al., 2023), ds004043 (Moerel et al., 2022), ds003885 (Shatek et al., 2021), ds004357 (Grootswagers et al., 2024), ds003825 (Grootswagers et al., 2022), ds004816 (Grootswagers et al., 2023a), ds004840 (Cordoba-Silva et al., 2023), ds005262 (Metwalli et al., 2024), ds004477 (Papastylianou et al., 2023), ds005273 (Esteban et al., 2024), ds004561 (Veillette et al., 2023), ds004951 (Haupt et al., 2024), ds004324 (Chacón and Wriessnegger, 2023), ds005095 (Zhozhikashvili et al., 2024), ds005509 (Shirazi et al., 2025), ds005505, ds005506, ds005507, ds005510, ds005511, ds005512, ds005514 (Shirazi et al., 2024b; Alexander et al., 2017) ds001787 (Delorme and Brandmeyer, 2024), ds003690 (Ribeiro and Castelo-Branco, 2021), ds004603 (Lowe et al., 2023), ds003969 (Delorme and Braboszcz, 2021), ds004147 (Hassall et al., 2024), ds003004 (Onton and Makeig, 2022), ds002721 (Daly et al., 2020), ds004152 (Hassall et al., 2022a) , ds005089 (Aguado-Lopez et al., 2024), ds004264 (Hassall et al., 2022b), ds004315 (Cavanagh and Jackson, 2022), ds004408 (Bialas et al., 2023), ds005121 (Siefert et al., 2024), ds003775 (Hatlestad-Hall et al., 2022), ds004572 (Kekecs and Farahzadi, 2024), ds002778 (Rockhill et al., 2020), ds003846 (Gehrke et al., 2024), ds004279 (Araya et al., 2023), ds004148 (Wang et al., 2022), ds004902 (Xiang et al., 2024), ds002680 (Delorme and Fabre-Thorpe, 2020), ds004284 (Veillette et al., 2022), ds004395 (Kahana et al., 2023),

ds005508 (Shirazi et al., 2024a), ds005697 (Li and Zhao, 2024), ds005620 (Bajwa1 et al., 2024), ds005594 (Taylor et al., 2024), ds005586 (Baykan and Schütz, 2024). OpenNeuro is under the Creative Commons CC0 license.

**Other sources:** NMT (Khan et al., 2022) under the Creative Commons Attribution License (CC BY), HMS (Ram et al., 2024) under the Attribution-NonCommercial 4.0 International (CC-BY-NC-4.0), SparrKULee (Accou et al., 2023) under the Attribution-Non Commercial 4.0 International (CC-BY-NC-4.0), Inria Large (Dreyer et al., 2023) the data on Zenodo being under the Creative Commons Attribution 4.0 International, THINGS2 (Gifford et al., 2022), under the CC-By Attribution 4.0 International license, TDBRAIN (Van Dijk et al., 2022), under the GPL-3.0 license, TUH (Obeid and Picone, 2016), freely available with registration required.

## C  Detailed results

This section presents detailed results on downstream tasks along with concise descriptions of the datasets.

### C.1  Emotion Recognition

**FACED** (Chen et al., 2023) We evaluate on the FACED dataset, which contains 32-channel EEG recordings (originally at 250 Hz, resampled to 200 Hz) from 123 subjects across nine emotion classes. The data is segmented into 10,332 samples of 10 seconds each. We follow the standard split: subjects 1–80 for training, 81–100 for validation, and 101–123 for testing.

Table 8: The results of different methods on emotion recognition (FACED, 9-class).

| Methods | Balanced Accuracy | Cohen's Kappa | Weighted F1 |
|---|---|---|---|
| EEGNet | $0.4090 \pm 0.0122$ | $0.3342 \pm 0.0251$ | $0.4124 \pm 0.0141$ |
| EEGConformer | $0.4559 \pm 0.0125$ | $0.3858 \pm 0.0186$ | $0.4514 \pm 0.0107$ |
| SPaRCNet | $0.4673 \pm 0.0155$ | $0.3978 \pm 0.0289$ | $0.4729 \pm 0.0133$ |
| ContraWR | $0.4887 \pm 0.0078$ | $0.4231 \pm 0.0151$ | $0.4884 \pm 0.0074$ |
| CNN-Transformer | $0.4697 \pm 0.0132$ | $0.4017 \pm 0.0168$ | $0.4720 \pm 0.0125$ |
| FFCL | $0.4673 \pm 0.0158$ | $0.3987 \pm 0.0383$ | $0.4699 \pm 0.0145$ |
| ST-Transformer | $0.4810 \pm 0.0079$ | $0.4137 \pm 0.0133$ | $0.4795 \pm 0.0096$ |
| BIOT | $0.5118 \pm 0.0118$ | $0.4476 \pm 0.0254$ | $0.5136 \pm 0.0112$ |
| LaBraM-Base | $0.5273 \pm 0.0107$ | $0.4698 \pm 0.0188$ | $0.5288 \pm 0.0102$ |
| CBraMod | $0.5509 \pm 0.0089$ | $0.5041 \pm 0.0122$ | $0.5618 \pm 0.0093$ |
| REVE-Base (ours) | $\mathbf{0.5646 \pm 0.0164}$ | $\mathbf{0.5080 \pm 0.0191}$ | $\mathbf{0.5659 \pm 0.0172}$ |

### C.2  Mental Disorder Diagnosis

**Mumtaz** (Mumtaz, 2016) We use the Mumtaz2016 dataset, which includes EEG recordings from 34 individuals with major depressive disorder (MDD) and 30 healthy controls, acquired from 19 electrodes (10–20 system) at 256 Hz. Only the eyes-open and eyes-closed sessions are used. Signals are band-pass filtered (0.3–75 Hz), notch filtered at 50 Hz, resampled to 200 Hz, and segmented into 7,143 samples of 5 seconds each. The split includes 24 MDD and 19 control subjects for training, 5 MDD and 4 controls for validation, and 5 MDD and 5 controls for testing. The dataset is under CC BY 4.0.

Table 9: The results of different methods on mental disorder diagnosis (Mumtaz2016, 2-class).

| Methods | Balanced Accuracy | AUC-PR | AUROC |
|---|---|---|---|
| EEGNet | $0.9232 \pm 0.0104$ | $0.9626 \pm 0.0095$ | $0.9639 \pm 0.0093$ |
| EEGConformer | $0.9308 \pm 0.0117$ | $0.9684 \pm 0.0105$ | $0.9702 \pm 0.0101$ |
| SPaRCNet | $0.9316 \pm 0.0095$ | $0.9754 \pm 0.0065$ | $0.9781 \pm 0.0083$ |
| ContraWR | $0.9195 \pm 0.0115$ | $0.9589 \pm 0.0102$ | $0.9621 \pm 0.0092$ |
| CNN-Transformer | $0.9305 \pm 0.0068$ | $0.9757 \pm 0.0074$ | $0.9742 \pm 0.0059$ |
| FFCL | $0.9314 \pm 0.0038$ | $0.9717 \pm 0.0021$ | $0.9753 \pm 0.0033$ |
| ST-Transformer | $0.9135 \pm 0.0103$ | $0.9578 \pm 0.0086$ | $0.9594 \pm 0.0059$ |
| BIOT | $0.9358 \pm 0.0052$ | $0.9736 \pm 0.0034$ | $0.9758 \pm 0.0042$ |
| LaBraM-Base | $0.9409 \pm 0.0079$ | $0.9798 \pm 0.0093$ | $0.9782 \pm 0.0057$ |
| CBraMod | $0.9560 \pm 0.0056$ | $0.9923 \pm 0.0032$ | $0.9921 \pm 0.0025$ |
| REVE-Base (ours) | $\mathbf{0.9644 \pm 0.0097}$ | $\mathbf{0.9961 \pm 0.0013}$ | $\mathbf{0.9957 \pm 0.0015}$ |

## C.3 Mental Stress Detection

**MAT** (Zyma et al., 2019) The MentalArithmetic dataset contains EEG recordings from 36 subjects, labeled as "with" or "without" mental stress depending on whether a mental arithmetic task was being performed. Signals were recorded from 20 electrodes (10–20 system) at 500 Hz, band-pass filtered (0.5–45 Hz), resampled to 200 Hz, and segmented into 1,707 samples of 5 seconds. Subjects 1–28 are used for training, 29–32 for validation, and 33–36 for testing. The MentalArithmetic dataset is under the Open Data Commons Attribution License v1.0.

Table 10: The results of different methods on mental stress detection (MAT, 2-class).

| Methods | Balanced Accuracy | AUC-PR | AUROC |
|---|---|---|---|
| EEGNet | $0.6770 \pm 0.0116$ | $0.5763 \pm 0.0102$ | $0.7321 \pm 0.0108$ |
| EEGConformer | $0.6805 \pm 0.0123$ | $0.5829 \pm 0.0134$ | $0.7424 \pm 0.0128$ |
| SPaRCNet | $0.6879 \pm 0.0107$ | $0.5825 \pm 0.0193$ | $0.7418 \pm 0.0132$ |
| ContraWR | $0.6631 \pm 0.0097$ | $0.5787 \pm 0.0164$ | $0.7332 \pm 0.0082$ |
| CNN-Transformer | $0.6779 \pm 0.0268$ | $0.5777 \pm 0.0285$ | $0.7258 \pm 0.0336$ |
| FFCL | $0.6798 \pm 0.0142$ | $0.5786 \pm 0.0266$ | $0.7330 \pm 0.0198$ |
| ST-Transformer | $0.6631 \pm 0.0173$ | $0.5672 \pm 0.0259$ | $0.7132 \pm 0.0174$ |
| BIOT | $0.6875 \pm 0.0186$ | $0.6004 \pm 0.0195$ | $0.7536 \pm 0.0144$ |
| LaBraM-Base | $0.6909 \pm 0.0125$ | $0.5999 \pm 0.0155$ | $0.7721 \pm 0.0093$ |
| CBraMod | $0.7256 \pm 0.0132$ | $0.6267 \pm 0.0099$ | $0.7905 \pm 0.0073$ |
| REVE-Base (ours) | $\mathbf{0.7660 \pm 0.0355}$ | $\mathbf{0.7470 \pm 0.0807}$ | $\mathbf{0.8450 \pm 0.0514}$ |

## C.4 Imagined Speech

**BCIC2020-3** (Jeong et al., 2022) BCIC2020-3 is an imagined speech EEG dataset from 15 subjects, recorded with 64 channels at 256 Hz while subjects silently imagined five phrases ("hello", "help me", "stop", "thank you", "yes") without any articulation. Each phrase has 80 trials per subject, totaling 6,000 3-second samples. The data is resampled to 200 Hz. The official split includes 60 trials per class for training, 10 for validation, and 10 for testing. BCIC2020-3 is under the Creative Commons Attribution No Derivatives license (CC BY-ND 4.0).

Table 11: The results of different methods on imagined speech classification (BCIC2020-3, 5-class).

| Methods | Balanced Accuracy | Cohen's Kappa | Weighted F1 |
|---|---|---|---|
| EEGNet | $0.4413 \pm 0.0096$ | $0.3016 \pm 0.0123$ | $0.4413 \pm 0.0102$ |
| EEGConformer | $0.4506 \pm 0.0133$ | $0.3133 \pm 0.0183$ | $0.4488 \pm 0.0154$ |
| SPaRCNet | $0.4426 \pm 0.0156$ | $0.3033 \pm 0.0233$ | $0.4420 \pm 0.0108$ |
| ContraWR | $0.4257 \pm 0.0162$ | $0.3078 \pm 0.0218$ | $0.4407 \pm 0.0182$ |
| CNN-Transformer | $0.4533 \pm 0.0092$ | $0.3166 \pm 0.0118$ | $0.4506 \pm 0.0127$ |
| FFCL | $0.4678 \pm 0.0197$ | $0.3301 \pm 0.0359$ | $0.4689 \pm 0.0205$ |
| ST-Transformer | $0.4126 \pm 0.0122$ | $0.2941 \pm 0.0159$ | $0.4247 \pm 0.0138$ |
| BIOT | $0.4920 \pm 0.0086$ | $0.3650 \pm 0.0176$ | $0.4917 \pm 0.0079$ |
| LaBraM-Base | $0.5060 \pm 0.0155$ | $0.3800 \pm 0.0242$ | $0.5054 \pm 0.0205$ |
| CBraMod | $0.5373 \pm 0.0108$ | $0.4216 \pm 0.0163$ | $0.5383 \pm 0.0096$ |
| REVE-Base (ours) | $\mathbf{0.5635 \pm 0.0123}$ | $\mathbf{0.4543 \pm 0.0154}$ | $\mathbf{0.5633 \pm 0.0124}$ |

## C.5 Motor Imagery Classification

**PhysioNet-MI** (Goldberger et al., 2000) is used for motor imagery classification. It contains recordings with 64 channels at a 160 Hz sampling rate and includes 4 classes: left fist, right fist, both fists, and feet. As in CBraMod, we select 4-second samples of the signals, resulting in 9,837 samples. Following CBraMod's protocol, subjects 1–70 are used for training, 71–89 for validation, and 90–109 for testing. We retain all subjects and use full 4-second windows to stay consistent with CBraMod. To handle lower sampling rates in some recordings, we load all data at 128 Hz (using a 64 Hz low-pass filter) before resampling to 200 Hz. Physionet-MI is under the Open Data Commons Attribution License v1.0.

**BCIC-IV-2a** (Tangermann et al., 2012) is also used for motor imagery classification. It contains EEG recordings from 9 subjects performing 4 motor imagery tasks: left hand, right hand, both feet, and tongue. Data were collected over 2 sessions with 22 electrodes at 250 Hz. Each session includes 288 trials (72 per task). We use the [2,6] second window from each trial, apply a 0.5–99.5 Hz band-pass filter, resample to 200 Hz, and apply Euclidean Alignment (He and Wu, 2019), proven to be effective on this task (El Ouahidi et al., 2024), resulting in 5184 4-second samples.

Table 12: The results of different methods on Motor Imagery classification.

| Methods | PhysioNet-MI, 4-class | | | BCIC-IV-2a, 4-class | | |
|---|---|---|---|---|---|---|
| | Balanced Accuracy | Cohen's Kappa | Weighted F1 | Balanced Accuracy | Cohen's Kappa | Weighted F1 |
| EEGNet | $0.5814 \pm 0.0125$ | $0.4468 \pm 0.0199$ | $0.5796 \pm 0.0115$ | $0.4482 \pm 0.0094$ | $0.2693 \pm 0.0121$ | $0.4226 \pm 0.0108$ |
| EEGConformer | $0.6049 \pm 0.0104$ | $0.4736 \pm 0.0171$ | $0.6062 \pm 0.0095$ | $0.4696 \pm 0.0106$ | $0.2924 \pm 0.0141$ | $0.4533 \pm 0.0128$ |
| SPaRCNet | $0.5932 \pm 0.0152$ | $0.4564 \pm 0.0234$ | $0.5937 \pm 0.0147$ | $0.4635 \pm 0.0117$ | $0.2847 \pm 0.0147$ | $0.4432 \pm 0.0126$ |
| ContraWR | $0.5892 \pm 0.0133$ | $0.4527 \pm 0.0248$ | $0.5918 \pm 0.0116$ | $0.4678 \pm 0.0125$ | $0.2905 \pm 0.0160$ | $0.4413 \pm 0.0142$ |
| (CNN-Transformer | $0.6053 \pm 0.0118$ | $0.4725 \pm 0.0223$ | $0.6041 \pm 0.0105$ | $0.4600 \pm 0.0108$ | $0.2800 \pm 0.0148$ | $0.4460 \pm 0.0114$ |
| FFCL | $0.5726 \pm 0.0092$ | $0.4323 \pm 0.0182$ | $0.5701 \pm 0.0079$ | $0.4470 \pm 0.0143$ | $0.2627 \pm 0.0176$ | $0.4238 \pm 0.0139$ |
| ST-Transformer | $0.6035 \pm 0.0081$ | $0.4712 \pm 0.0199$ | $0.6053 \pm 0.0075$ | $0.4575 \pm 0.0145$ | $0.2733 \pm 0.0198$ | $0.4471 \pm 0.0142$ |
| BIOT | $0.6153 \pm 0.0154$ | $0.4875 \pm 0.0272$ | $0.6158 \pm 0.0197$ | $0.4748 \pm 0.0093$ | $0.2997 \pm 0.0139$ | $0.4607 \pm 0.0125$ |
| LaBraM-Base | $0.6173 \pm 0.0122$ | $0.4912 \pm 0.0192$ | $0.6177 \pm 0.0141$ | $0.4869 \pm 0.0085$ | $0.3159 \pm 0.0154$ | $0.4758 \pm 0.0103$ |
| CBraMod | $0.6417 \pm 0.0091$ | $0.5222 \pm 0.0169$ | $0.6427 \pm 0.0100$ | $0.5138 \pm 0.0066$ | $0.3518 \pm 0.0094$ | $0.4984 \pm 0.0085$ |
| REVE-Base (ours) | $\mathbf{0.6480 \pm 0.0140}$ | $\mathbf{0.5306 \pm 0.0187}$ | $\mathbf{0.6484 \pm 0.0170}$ | $\mathbf{0.6396 \pm 0.0095}$ | $\mathbf{0.5194 \pm 0.0126}$ | $\mathbf{0.6339 \pm 0.0110}$ |

## C.6 Sleep Staging

**ISRUC** (Khalighi et al., 2016) We use the sleep staging task on the ISRUC dataset (Subgroup 1), which contains PSG recordings from 100 subjects. Only EEG signals are used (6 channels, sampled at 200 Hz), segmented into 89,240 30-second epochs, each labeled with one of five sleep stages following AASM standards. Subjects 1–80 are used for training, 81–90 for validation, and 91–100 for testing. As in prior work, the task is framed as a sequence-to-sequence classification problem, using sequences of 20 consecutive epochs to model stage transitions. ISRUC is freely accessible online.

Table 13: The results of different methods on sleep staging (ISRUC, 5-class). [*] In the baseline code, a chin electrode might have been used instead of an EEG one; REVE results are reported without it.

| Methods | Balanced Accuracy | Cohen's Kappa | Weighted F1 |
|---|---|---|---|
| EEGNet | $0.7154 \pm 0.0121$ | $0.7040 \pm 0.0173$ | $0.7513 \pm 0.0124$ |
| EEGConformer | $0.7400 \pm 0.0133$ | $0.7143 \pm 0.0162$ | $0.7634 \pm 0.0151$ |
| SPaRCNet | $0.7487 \pm 0.0075$ | $0.7097 \pm 0.0132$ | $0.7624 \pm 0.0092$ |
| ContraWR | $0.7402 \pm 0.0126$ | $0.7178 \pm 0.0156$ | $0.7610 \pm 0.0137$ |
| CNN-Transformer | $0.7363 \pm 0.0087$ | $0.7129 \pm 0.0121$ | $0.7719 \pm 0.0105$ |
| FFCL | $0.7277 \pm 0.0182$ | $0.7016 \pm 0.0291$ | $0.7614 \pm 0.0197$ |
| ST-Transformer | $0.7381 \pm 0.0205$ | $0.7013 \pm 0.0352$ | $0.7681 \pm 0.0175$ |
| DeepSleepNet | $0.7419 \pm 0.0144$ | $0.7036 \pm 0.0241$ | $0.7643 \pm 0.0122$ |
| USleep | $0.7586 \pm 0.0116$ | $0.7209 \pm 0.0143$ | $0.7805 \pm 0.0105$ |
| BIOT | $0.7527 \pm 0.0121$ | $0.7192 \pm 0.0231$ | $0.7790 \pm 0.0146$ |
| LaBraM-Base | $0.7633 \pm 0.0102$ | $0.7231 \pm 0.0182$ | $0.7810 \pm 0.0133$ |
| CBraMod | $\mathbf{0.7865 \pm 0.0110}$ | $0.7442 \pm 0.0152$ | $\mathbf{0.8011 \pm 0.0099}$ |
| REVE-Base[*] | $0.7819 \pm 0.0078$ | $\mathbf{0.7500 \pm 0.0156}$ | $0.8005 \pm 0.0135$ |

**HMC** (Alvarez-Estevez and Rijsman, 2021). The Haaglanden Medisch Centrum (HMC) Sleep Staging Database is a sleep stage detection dataset, consisting of 151 full-night polysomnographic (PSG) recordings collected from patients referred for sleep studies. The data includes EEG, EOG, EMG, and ECG channels, with a sampling rate of 256 Hz, and annotations for five sleep stages (Wake, N1, N2, N3, REM) manually scored by trained sleep technicians. HMC is under the Creative Commons Attribution 4.0 International Public License.

Table 14: The results of different methods on sleep staging (HMC, 5-class).

| Methods | Balanced Accuracy | Cohen's Kappa | Weighted F1 |
|---|---|---|---|
| SPaRCNet | $0.4756 \pm 0.1109$ | $0.3147 \pm 0.1315$ | $0.4108 \pm 0.1310$ |
| ContraWR | $0.4242 \pm 0.0541$ | $0.2340 \pm 0.0554$ | $0.2987 \pm 0.0288$ |
| CNN-Transformer | $0.6573 \pm 0.0141$ | $0.5961 \pm 0.0105$ | $0.6896 \pm 0.0065$ |
| FFCL | $0.4427 \pm 0.0702$ | $0.2542 \pm 0.0654$ | $0.2902 \pm 0.0485$ |
| ST-Transformer | $0.2559 \pm 0.0141$ | $0.0503 \pm 0.0183$ | $0.1428 \pm 0.0122$ |
| BIOT | $0.6862 \pm 0.0041$ | $0.6295 \pm 0.0113$ | $0.7091 \pm 0.0147$ |
| LaBraM-Base | $0.7286 \pm 0.0101$ | $0.6812 \pm 0.0073$ | $0.7554 \pm 0.0024$ |
| REVE-Base | $\mathbf{0.7401 \pm 0.0075}$ | $\mathbf{0.6982 \pm 0.0078}$ | $\mathbf{0.7638 \pm 0.0074}$ |

## C.7 Event Type Classification

**TUEV** (Obeid and Picone, 2016) is an EEG dataset with six annotated classes: spike and sharp wave, generalized periodic epileptiform discharges, periodic lateralized epileptiform discharges, eye movement, artifact, and background. The recordings use 23 channels at a 256 Hz sampling rate. For consistency with CBraMod, BIOT, and LaBraM, we used BIOT's processing scripts which preprocess the dataset using 16 common bipolar montage channels in the 10-20 system, apply a 0.3–75 Hz band-pass filter, remove power line noise with a 60 Hz notch filter, and resample to 200 Hz. The dataset is split into 112,491 5-second samples. We follow the original training-test split and further divide the training set into 80% training and 20% validation, matching BIOT setting. To provide our model with the electrode positions, we used the average position of each bipolar montage. TUEV is part of the TUH dataset, which is freely available with registration required.

Table 15: The results of different methods on event type classification (TUEV, 6-class).

| Methods | Balanced Accuracy | Cohen's Kappa | Weighted F1 |
|---|---|---|---|
| EEGNet | 0.3876 ± 0.0143 | 0.3577 ± 0.0155 | 0.6539 ± 0.0120 |
| EEGConformer | 0.4074 ± 0.0164 | 0.3967 ± 0.0195 | 0.6983 ± 0.0152 |
| SPaRCNet | 0.4161 ± 0.0262 | 0.4233 ± 0.0181 | 0.7024 ± 0.0104 |
| ContraWR | 0.4384 ± 0.0349 | 0.3912 ± 0.0237 | 0.6893 ± 0.0136 |
| CNN-Transformer | 0.4087 ± 0.0161 | 0.3815 ± 0.0134 | 0.6854 ± 0.0293 |
| FFCL | 0.3979 ± 0.0104 | 0.3732 ± 0.0188 | 0.6783 ± 0.0120 |
| ST-Transformer | 0.3984 ± 0.0228 | 0.3765 ± 0.0306 | 0.6823 ± 0.0190 |
| BIOT | 0.5281 ± 0.0225 | 0.5273 ± 0.0249 | 0.7492 ± 0.0082 |
| LaBraM-Base | 0.6409 ± 0.0065 | 0.6637 ± 0.0093 | 0.8312 ± 0.0052 |
| LaBraM-Large | 0.6581 ± 0.0156 | 0.6622 ± 0.0136 | 0.8315 ± 0.0040 |
| LaBraM-Huge | 0.6616 ± 0.0170 | 0.6745 ± 0.0195 | 0.8329 ± 0.0086 |
| CBraMod | 0.6671 ± 0.0107 | 0.6772 ± 0.0096 | 0.8342 ± 0.0064 |
| REVE-Base (ours) | **0.6759** ± 0.0229 | **0.6783** ± 0.0199 | **0.8451** ± 0.0129 |

## C.8 Abnormal Detection

**TUAB** (Obeid and Picone, 2016) is used for abnormal EEG detection, where recordings are labeled as normal or abnormal. It shares the same 23-channel, 256 Hz format as TUEV. The dataset is split into 409,455 10-second samples for binary classification. We follow the provided training-test split and apply an 80%-20% training-validation split, consistent with BIOT. We resampled at 200 Hz, band-pass at 0.5-99.5 Hz, and directly used all channels and their positions. TUAB is part of the TUH dataset, which is freely available with registration required.

Table 16: The results of different methods on abnormal detection (TUAB, 2-class).

| Methods | Balanced Accuracy | AUC-PR | AUROC |
|---|---|---|---|
| EEGNet | 0.7642 ± 0.0036 | 0.8299 ± 0.0043 | 0.8412 ± 0.0031 |
| EEGConformer | 0.7758 ± 0.0049 | 0.8427 ± 0.0054 | 0.8445 ± 0.0038 |
| SPaRCNet | 0.7896 ± 0.0018 | 0.8414 ± 0.0018 | 0.8676 ± 0.0012 |
| ContraWR | 0.7746 ± 0.0041 | 0.8421 ± 0.0104 | 0.8456 ± 0.0074 |
| CNN-Transformer | 0.7777 ± 0.0022 | 0.8433 ± 0.0039 | 0.8461 ± 0.0013 |
| FFCL | 0.7848 ± 0.0038 | 0.8448 ± 0.0065 | 0.8569 ± 0.0051 |
| ST-Transformer | 0.7966 ± 0.0023 | 0.8521 ± 0.0026 | 0.8707 ± 0.0019 |
| BIOT | 0.7959 ± 0.0057 | 0.8792 ± 0.0023 | 0.8815 ± 0.0043 |
| LaBraM-Base | 0.8140 ± 0.0019 | 0.8965 ± 0.0016 | 0.9022 ± 0.0009 |
| LaBraM-Large | 0.8226 ± 0.0015 | 0.9130 ± 0.0005 | 0.9127 ± 0.0005 |
| LaBraM-Huge | 0.8258 ± 0.0011 | 0.9204 ± 0.0011 | 0.9162 ± 0.0016 |
| CBraMod | 0.8289 ± 0.0022 | 0.9258 ± 0.0008 | 0.9227 ± 0.0011 |
| REVE-Base (ours) | **0.8315** ±0.0014 | **0.9281** ±0.0009 | **0.9245** ±0.0013 |

## D Ablation on the SSL Method

The final pretraining hyperparameters were selected based on a series of ablation studies, the results of which are presented in this section.

Table 17 reports the impact of the secondary pretraining loss on eight downstream tasks using REVE-Small, evaluated under frozen-backbone, linear probing (LP), and full fine-tuning (FT) settings. Results obtained with both losses are compared to those using only the primary loss. The secondary loss consistently improves performance across nearly all datasets, enhancing results in both LP and FT settings, while its removal leads to a substantial drop, underscoring its importance for the model to produce strong embeddings.

The results in Table 18 show that a block masking ratio of 55% yields the best overall performance, providing stable results across both fine-tuned and frozen settings and eight datasets (Mumtaz, TUAB, ISRUC, HMC, BCIC2020-3, TUEV, PhysioNetMI, and Faced). In contrast, random masking

Table 17: Effect of 2nd loss during pretraining and finetuning. The reported metric is balanced accuracy. Best results per dataset are in bold.

| | LP | | FT | |
| Dataset | No 2nd loss | + 2nd loss | No 2nd loss | + 2nd loss |
|---|---|---|---|---|
| Mumtaz | $0.818 \pm 0.043$ | $0.920 \pm 0.018$ | $0.818 \pm 0.043$ | $\mathbf{0.922 \pm 0.018}$ |
| TUAB | $0.797 \pm 0.004$ | $0.802 \pm 0.005$ | $0.803 \pm 0.003$ | $\mathbf{0.810 \pm 0.005}$ |
| ISRUC | $0.699 \pm 0.006$ | $0.625 \pm 0.003$ | $\mathbf{0.777 \pm 0.002}$ | $0.770 \pm 0.002$ |
| HMC | $0.598 \pm 0.008$ | $0.591 \pm 0.005$ | $0.713 \pm 0.011$ | $\mathbf{0.723 \pm 0.005}$ |
| BCIC2020-3 | $0.234 \pm 0.009$ | $0.237 \pm 0.008$ | $0.390 \pm 0.017$ | $\mathbf{0.481 \pm 0.008}$ |
| TUEV | $0.442 \pm 0.060$ | $0.520 \pm 0.005$ | $0.533 \pm 0.024$ | $\mathbf{0.623 \pm 0.011}$ |
| PhysioNetMI | $0.379 \pm 0.058$ | $0.533 \pm 0.019$ | $0.563 \pm 0.011$ | $\mathbf{0.583 \pm 0.009}$ |
| Faced | $0.220 \pm 0.008$ | $0.233 \pm 0.004$ | $0.302 \pm 0.016$ | $\mathbf{0.410 \pm 0.004}$ |
| Avg. | 0.523 | 0.558 | 0.612 | **0.665** |

Table 18: Performance comparison across different masking ratios (0.25, 0.55, 0.75) between block masking strategy and random masking, evaluated for full fine-tuning versus frozen embeddings. We display the average balanced accuracy on the small model over eight downstream tasks.

| | Frozen | | Full Fine-Tuning | |
| Masking Ratio | Random | Block | Random | Block |
|---|---|---|---|---|
| 0.25 | 0.523 | 0.513 | 0.612 | 0.602 |
| 0.55 | 0.550 | **0.558** | 0.643 | **0.665** |
| 0.75 | 0.519 | 0.546 | 0.606 | 0.655 |

favors smaller ratios (25%), but its unstructured nature leads to highly redundant inputs, making the reconstruction task artificially easier. These findings align with ablation results reported in Cbramod, Labram, and BIOT.

Table 19: Ablation study on PhysioNetMI and Mental Arithmetic datasets. The reported metric is balanced accuracy, with the average computed across both tasks, with the Base model.
[*]Note that the learnable positional encoding matches the baseline, but does not allow for the extension to larger time windows or unseen spatial configurations.

| Ablated component | PhysionetMI | Mental Arithmetic | Average |
|---|---|---|---|
| Learnable PE[*] | $\mathbf{0.650} \pm 0.0113$ | $0.752 \pm 0.0421$ | $0.701 \pm 0.0218$ |
| MLP4D | $0.637 \pm 0.0056$ | $0.717 \pm 0.0425$ | $0.677 \pm 0.0214$ |
| Position noise | $0.628 \pm 0.0084$ | $0.692 \pm 0.0665$ | $0.660 \pm 0.0335$ |
| Dropout block masking | $0.645 \pm 0.0155$ | $0.678 \pm 0.0521$ | $0.662 \pm 0.0272$ |
| Temporal block masking | $0.646 \pm 0.0155$ | $0.723 \pm 0.0422$ | $0.685 \pm 0.0225$ |
| Base Performance | $0.6480 \pm 0.0140$ | $\mathbf{0.7660 \pm 0.0355}$ | $\mathbf{0.707 \pm 0.0191}$ |

Table 19 presents an ablation study on two downstream tasks to assess the contribution of each component in our SSL pipeline. All components appear to contribute positively to performance. The "Learnable PE" line is not a true ablation, but rather a variant using learnable positional embeddings, where a separate embedding is learned for each electrode and time index observed during pretraining. Although this approach performs well, it is limited to the spatial and temporal configurations seen during training (approximately 400 unique electrode names, over 10-second windows) and does not generalize to longer sequences or unseen electrode layouts, unlike REVE's 4D positional encoding.

Table 20 presents an ablation study on the choice of activation and normalization functions, an important design factor in transformer-based foundation models. We compare GEGLU + RMSNorm, GELU + RMSNorm, and GEGLU + LayerNorm configurations during pretraining, and report downstream performance after fine-tuning on three datasets using the REVE-Small model.

Table 20: Ablation study on activation functions and normalization layers (GEGLU vs. GELU, RMSNorm vs. LayerNorm). We report downstream balanced accuracy after pretraining the REVE-Small model with each configuration.

| Dataset | GEGLU + RMSNorm | GELU + RMSNorm | GEGLU + LayerNorm |
|---|---|---|---|
| BCIC-IV-2a | **0.581±0.012** | 0.560 ± 0.018 | 0.537 ± 0.018 |
| TUEV | **0.623 ± 0.011** | 0.592± 0.010 | 0.577±0.034 |
| PhysioNetMI | 0.583 ± 0.009 | **0.586 ±0.009** | 0.559 ±0.007 |
| Avg. | **0.596** | 0.579 | 0.558 |

The GEGLU + RMSNorm combination achieves the best average performance (0.596), outperforming the others on BCIC-IV-2a and TUEV. GELU + RMSNorm performs similarly but only leads on PhysioNetMI. In contrast, GEGLU + LayerNorm consistently underperforms, highlighting the effectiveness of RMSNorm over LayerNorm and the benefits of gated activations like GEGLU in this context.

# E    Additional results

This section presents supplementary experiments that further support the main results, focusing on few-shot performance and evaluation under reduced-electrode configurations.

## E.1    Sparse setups

Table 21: Performance of REVE-Base under sparse input configurations. Balanced accuracy is reported for PhysionetMI (Left–Right) and imagined speech tasks as the number of EEG channels is progressively reduced.

| Channels | PhysionetMI L-R | Speech |
|---|---|---|
| 64 | 0.824 ± 0.008 | 0.565 ±0.016 |
| 32 | 0.808 ± 0.007 | 0.490 ± 0.094 |
| 16 | 0.787 ± 0.008 | 0.469 ±0.014 |
| 8 | 0.781 ± 0.006 | 0.294 ± 0.063 |
| 4 | 0.728 ± 0.009 | 0.258 ± 0.019 |
| 2 | 0.700 ± 0.025 | 0.228 ± 0.006 |
| 1 | 0.660 ±0.019 | 0.209 ± 0.008 |

Table 21 reports REVE-Base's performance under increasingly sparse input configurations. On the Physionet MI L-R task, accuracy degrades gracefully from 0.824 with 64 channels to 0.660 with a single channel, demonstrating robustness to reduced spatial coverage. In contrast, the imagined speech task is more sensitive to channel sparsity, with performance dropping from 0.565 to 0.258 with four channels and 0.209 with one, close to random chance. These results confirm that while REVE generalizes well under limited input, tasks requiring broad spatial information remain more challenging.

## E.2    Few-shot experiments

We conducted few-shot (FS) experiments to simulate realistic BCI usage scenarios. Tasks were constructed from the BCI IV-2a dataset using two motor imagery classes (Left–Right). For each subject, multiple inductive FS runs were performed. In each run, $N$ labeled samples per class ("shots") were randomly selected within a session for training, while the remaining samples from both sessions were used for evaluation.

Classification was done using a Nearest Class Mean (NCM) classifier. Each configuration was repeated 20 times per subject, and we report the average balanced accuracy across subjects and runs. We evaluated two configurations of REVE-Base:

- REVE-Base (PT): directly after self-supervised pretraining, with no further supervised adaptation.
- REVE-Base (XFT): after cross-dataset fine-tuning on multiple labeled Left–Right MI datasets ((Schirrmeister et al., 2017), (Cho et al., 2017), (Goldberger et al., 2000), (Lee et al., 2019), (Yi et al., 2014)). REVE's 4D positional encoding enables joint training across diverse electrode configurations without requiring channel alignment or selection.

Table 22: Few-shot performance of REVE-Base on BCI IV-2a dataset

| N-shots | 1 | 2 | 5 | 10 | 20 |
|---|---|---|---|---|---|
| REVE-Base (PT) | $0.588 \pm 1.45$ | $0.601 \pm 0.001$ | $0.652 \pm 0.013$ | $0.688 \pm 0.010$ | $0.723 \pm 0.010$ |
| REVE-Base (XFT) | $0.605 \pm 1.12$ | $0.645 \pm 0.009$ | $0.705 \pm 0.009$ | $0.768 \pm 0.009$ | $0.817 \pm 0.004$ |

Table 22 shows that REVE-Base achieves competitive accuracy even without supervised adaptation, demonstrating that its pretrained embeddings can be effectively leveraged for downstream BCI tasks. After cross-dataset fine-tuning, performance improves consistently across all shot counts, with gains reaching +10% at 20 shots. This indicates that REVE transfers well across subjects and datasets, while benefiting from minimal supervised adaptation. Such generalization is uncommon among BCI embedding models, which typically require task or subject-specific retraining.

# F   Experiment details

## F.1   Compute resources

We include details about the compute nodes that were used for pretraining.

- Compute Type: GPU-accelerated nodes
- GPU Model: NVIDIA A100
- CPU Model: Intel Cascade Lake SP 6248
- CPU Cores per Node: 40 cores
- Total Memory per Node: 192 GB
- Storage: Access to a shared full-flash parallel file system based on IBM Spectrum Scale
- Job Scheduler: Slurm

We also estimate the number of floating-point operations (FLOPs) required to train the REVE-Base model, following the formulation from Chowdhery et al. (2023):

$$\tau = \frac{D \cdot (6N + 12LHQT)}{P \cdot \eta}$$

where $\tau$ denotes the training time (in seconds), $D = 60\text{k} \times 3600 \times 1.1 \times 68 \times 17$ is the total number of tokens seen during pretraining (corresponding to 60k hours of EEG, an overlap coefficient of 1.1, 68 average channels, and 17 epochs), $N = 72\text{M}$ is the number of model parameters, $L = 23$ the number of encoder-decoder layers, $H = 8$ the number of attention heads, $Q = 64$ the head dimension, and $T = 68 \times 11$ the average number of tokens per sequence (channels × patches).

The peak throughput is $P = 312$ TFLOPs at half precision, achievable on A100 GPUs, and the model FLOPs utilization is set to $\eta = 0.5$ (50%).

This configuration yields an estimated 260 A100 GPU hours for a single pretraining run. The formula can be directly adapted for other model sizes or hardware configurations.

## F.2   Use of Existing Assets

We used Python (Python Software Foundation License), and some associated libraries for the implementation:

1. PyTorch (BSD-3 License)
2. NumPy (NumPy license)
3. scikit-learn (BSD license)
4. Pandas (BSD 3-Clause License)
5. Hugging Face's Accelerate (Apache License 2.0)

