# OpenReview forum: "REVE: A Foundation Model for EEG - Adapting to Any Setup with Large-Scale Pretraining on 25,000 Subjects"
_NeurIPS.cc/2025/Conference — NeurIPS 2025 poster_

### Official Review · Reviewer_aUp9 · 2025-06-30

**Clarity:** 3
**Significance:** 3
**Originality:** 2
**Rating:** 4
**Confidence:** 5

**Summary:**

This paper addresses the limited generalization of existing EEG foundation models across diverse datasets, especially under linear probing. The authors propose REVE, a new pretrained model with a novel 4D positional encoding scheme that enables flexible handling of variable-length signals and electrode configurations. Trained on a large-scale EEG corpus, REVE achieves strong performance across multiple downstream tasks with minimal fine-tuning, offering a promising step toward more robust and transferable EEG representations.

**Questions:**

- The paper adopts a Transformer-based architecture with both encoder and decoder, but it’s not entirely clear why the decoder is needed. From what I could tell, the model doesn’t seem to involve any autoregressive training — so what role does the decoder play in the overall design? A clearer explanation would help readers understand its purpose.
- The authors borrow several training techniques from large language models, such as RMSNorm and GEGLU activation. While these components are well-established in NLP, it’s less clear whether they bring real benefits to EEG foundation model training compared to more traditional choices like LayerNorm or GELU. Some ablation or discussion on this would be helpful.

**Ethical Concerns:**

["NO or VERY MINOR ethics concerns only"]

**Final Justification:**

The authors have adequately addressed most of my concerns regarding linear probing results, architectural clarifications, and ablation studies. I maintain my positive score and look forward to the open-sourced code and pretrained weights contributing to and inspiring the EEG research community.

**Limitations:**

Yes

**Paper Formatting Concerns:**

The paper uses only narrative-style citations throughout the manuscript, without distinguishing between narrative and parenthetical styles. A clearer and more consistent use of both types — depending on sentence structure and emphasis — would improve readability and align the writing with standard academic conventions.

---
### Parenthetical vs. Narrative Citations: A Simple Comparison

| Feature                     | Parenthetical Citation             | Narrative Citation                  |
|----------------------------|------------------------------------|-------------------------------------|
| **Format**                 | (Author et al., Year)              | Author et al. (Year)                |
| **Use Case**               | When citing a fact or idea without emphasizing the author | When highlighting the author or their work |
| **Example 1**              | This has been observed before (Zhang et al., 2020). | Zhang et al. (2020) observed this effect. |
| **Example 2**              | EEG modeling has improved in recent years (Wang & Lee, 2021). | Wang and Lee (2021) introduced a new approach to EEG modeling. |
| **Tone / Emphasis**        | Neutral, background reference      | Direct, highlights the authors      |

**Quality:**

3

**Strengths And Weaknesses:**

#### Strengths

- The scale of the pretraining dataset is truly impressive, covering over 60,000 hours of EEG data from 92 different datasets and 25,000 subjects. This kind of breadth has not been seen before in the field of EEG foundation models.

- The proposed 4D positional encoding scheme introduces a fresh idea for handling EEG signals with varying electrode configurations and lengths. It’s a thoughtful design that could help future models better adapt to real-world EEG setups.

- The authors evaluated their model across a wide range of downstream tasks — including motor imagery, seizure detection, sleep staging, emotion recognition, and more — showing both broad coverage and solid performance across the board.

- The results are consistently strong, often outperforming existing methods by a meaningful margin. Especially under linear probing or minimal fine-tuning settings, REVE still delivers competitive accuracy, which speaks well of its generalization capability.


#### Weaknesses

- Although the paper claims strong performance under linear probing, this is mostly supported by experiments on PhysioNet-MI. That’s just one dataset, and it’s hard to draw general conclusions from such limited evidence. More cross-dataset comparisons would make this claim much stronger.

- Looking at Table 17, it seems like the secondary loss might actually be the key factor behind REVE’s impressive linear probing performance. However, the authors don’t explain what this loss does or why it helps. It feels underemphasized, especially given how important it appears to be. I’d strongly suggest expanding on this part and running more ablation studies to back it up.

- While the empirical results are convincing, the paper lacks deeper analysis. There’s no visualization or interpretability study showing what the model learns. Also, the advantage of the 4D positional encoding over standard approaches isn’t clearly demonstrated through ablation experiments.

- Some practical details are missing — for example, how many devices were used during training, or how long the pretraining actually took. These are important for reproducibility and understanding the computational cost. There are also some citation formatting issues that should be cleaned up.

---

> ### Author Rebuttal · Authors · 2025-07-31
>
> We thank reviewer `aUp9` for their thoughtful and constructive feedback, as well as for recognizing the breadth, generalization capabilities, and novelty of REVE. We address concerns about linear probing results, terminology, the secondary loss, interpretability results, the performance of our 4D positional encoding, the compute cost of our pretraining runs, and the citation formatting.
>
> ## Linear probing results on more datasets
>
> > Although the paper claims strong performance under linear probing, this is mostly supported by experiments on PhysioNet-MI. That’s just one dataset, and it’s hard to draw general conclusions from such limited evidence. More cross-dataset comparisons would make this claim much stronger.
>
> We agree that broader linear probing evaluation is critical and could improve the impact of the paper. In response, we extended linear probing experiments to all downstream datasets and combined existing results in Table 16 and new experiments per reviewer `rHPq`’s simultaneous request. This results in table `rHPq.1`, from which we indicate the average performance across all datasets:
>
> |Dataset|REVE-B (Pool)|REVE-B (No Pool)|REVE-L (Pool)|REVE-L (No Pool)|CBraMod (Pool)|CBraMod (No Pool)|
> |-|-|-|-|-|-|-|
> |Avg.|0.586|0.609|0.623|**0.654**|0.373|0.501|
>
> Table `aUp9.1`: Average linear probing results on downstream tasks for REVE and CBraMod. Best results are highlighted in bold.
>
> For more context on the additional results, we reproduced CBraMod (Wang et al., 2025) using the official code and checkpoint, and ran it under the same linear probing setup as theirs across those datasets (with and without pooling).
>
> These results confirm that REVE consistently outperforms CBraMod under frozen encoder settings, supporting its generalization under minimal fine-tuning. We will move these new results into the main body for visibility.
>
> ## Decoder terminology
>
> > The paper adopts a Transformer-based architecture with both encoder and decoder, but it’s not entirely clear why the decoder is needed. From what I could tell, the model doesn’t seem to involve any autoregressive training — so what role does the decoder play in the overall design? A clearer explanation would help readers understand its purpose.
>
> We thank the reviewer for the helpful comment regarding terminology.
> In the manuscript, we use the terms "encoder" and "decoder" in the context of masked auto-encoders, which follow an encoder-decoder structure. For readers more familiar with the Transformer-specific terminology, this can indeed be interpreted as referring to non-causal and causal attention patterns, respectively. To avoid any ambiguity, we will clarify in the revision that all Transformer blocks used in our model are non-causal, corresponding to standard "encoder" blocks. We will also explicitly state that no autoregressive training is involved.
>
> ## Secondary loss
>
> > Looking at Table 17, it seems like the secondary loss might actually be the key factor behind REVE’s impressive linear probing performance. However, the authors don’t explain what this loss does or why it helps. It feels underemphasized, especially given how important it appears to be. I’d strongly suggest expanding on this part and running more ablation studies to back it up.
>
> We recognize that including more results on the secondary loss could improve the paper. Following the simultaneous request of reviewer `G86N`, we ran the ablation studies on the full downstream task suite, and will also update table 17 with the complete results.
>
> In the Table below, we report the balanced accuracy using REVE-Small
>
> |Dataset|no 2nd loss (LP)|no 2nd loss (FT)|2nd loss (LP)|2nd loss (FT)|
> |-|-|-|-|-|
> |Mumtaz|0.818 ± 0.043|0.818 ± 0.043|0.920 ± 0.018|**0.922 ± 0.018**|
> |TUAB|0.797 ± 0.004|0.803 ± 0.003|0.802 ± 0.005|**0.810 ± 0.005**|
> |ISRUC|0.699 ± 0.006|**0.777 ± 0.002**|0.625 ± 0.003|0.770 ± 0.002|
> |HMC|0.598 ± 0.008|0.713 ± 0.011|0.591 ± 0.005|**0.723 ± 0.005**|
> |BCIC2020-3|0.234 ± 0.009|0.390 ± 0.017|0.237 ± 0.008|**0.481 ± 0.008**|
> |TUEV|0.442 ± 0.060|0.533 ± 0.024|0.520 ± 0.005|**0.623 ± 0.011**|
> |PhysioNetMI|0.379 ± 0.058|0.563 ± 0.011|0.533 ± 0.019|**0.583 ± 0.009**|
> |Faced|0.220 ± 0.008|0.302 ± 0.016|0.233 ± 0.004|**0.410 ± 0.004**|
> |Avg.|0.523|0.612|0.558|**0.665**|
>
> Table `aUp9.2`: Secondary loss ablation results on REVE-Small
>
> The extended analysis supports our claim, showing improved average downstream performance in both LP and FT scenarios. For further intuition on the secondary loss, we refer to our response to reviewer `G86N`. The additional results support our claims in this section and highlight the positive impact of the secondary loss on the quality of the pooled pretrained embeddings.
>
> ## Interpretability results
>
> > While the empirical results are convincing, the paper lacks deeper analysis. There’s no visualization or interpretability study showing what the model learns. Also, the advantage of the 4D positional encoding over standard approaches isn’t clearly demonstrated through ablation experiments.
>
> We acknowledge the value of deeper analysis and interpretability results. We point out that we already provided initial experiments on interpretability at submission time on the public repository. We provide attention maps for Motor Imagery and for Imagined speech decoding in the figs/ folder, along with the code to reproduce them. Per NeurIPS policy, we can not add new interpretability figures beyond what was already provided. We strongly consider a focus on interpretability for future work.
>
> ## 4D positional encoding
>
> We would like to clarify the advantage of the 4D PE over fixed learned embeddings. The main benefit of the 4D positional encoding is handling heterogeneous electrode configurations unknown at training time, rather than strictly outperforming learned embeddings in raw metrics. This is the key interpretation behind Table 19.
>
> ## Compute cost
>
> > Some practical details are missing — for example, how many devices were used during training, or how long the pretraining actually took. These are important for reproducibility and understanding the computational cost. There are also some citation formatting issues that should be cleaned up.
>
> We agree that reporting the training time and necessary compute is important. We will include the following calculation in the final manuscript.
>
> We first derive the number of floating point operations (FLOPs) required for the training of REVE Base using the following formula derived from appendix B from PALM (Chowdhery et al.,  2023).
>
> $$
> \tau =  \frac{D \cdot (6N+12LHQT)}{P \cdot \eta}
> $$
> where:
> - $\tau$ is the training time in seconds
> - $D = 60k * 3600 * 1.1 * 68 * 17$ is the total number of token seen during pretraining (60k hours, 1.1 for the patch overlap coefficient, 68 channels on average and 17 epochs)
> - $N = 72M$ parameters for the base model
> - $L = 23$ layers in the encoder + decoder
> - $H = 8$ the number of attention heads
> - $Q = 64$ the head dimension
> - $T = 68 * 11$ the average number of tokens per sequence (number of channels by 11 patches)
> - $P = 312T$ the peak throughput in TFLOPs at half precision, achievable on A100 GPUs
> - $\eta = 0.5$ the model flops utilization (MFU), a measure of efficiency of the training run, which we set to 50% as a conservative estimate.
>
> This results in 260 A100 GPU hours.
>
> ## Ablation on normalization and activation layers
>
> > The authors borrow several training techniques from large language models, such as RMSNorm and GEGLU activation. While these components are well-established in NLP, it’s less clear whether they bring real benefits to EEG foundation model training compared to more traditional choices like LayerNorm or GELU.
>
> While we argue that many components of our model, including the Transformer backbone itself, are modality agnostic, we agree that those claims could be backed up by more empirical evidence. We acknowledge that many architectural choices are taken from the well established NLP literature, such as using GEGLU as an activation function. To back up our architectural choices, we ran ablations with LayerNorm and GELU, the results of which will be included in the final manuscript.
>
> RMSNorm (Zhang et al., 2019) is empirically modality agnostic, as shown in the original paper where the method is applied on both text and image processing. To further back up this claim, we ran ablation studies with LayerNorm and GELU.
>
> |Dataset|GEGLU + RMSNorm|GELU + RMSNorm|GEGLU + LayerNorm|
> |-|-|-|-|
> |BCIC-IV-2a|**0.581±0.012**|0.560 ± 0.018|0.537 ± 0.018|
> |TUEV|**0.623 ± 0.011**|0.592± 0.010|0.577±0.034|
> |PhysioNetMI|0.583 ± 0.009|**0.586 ±0.009**|0.559 ±0.007|
> |Avg.|**0.596 ± 0.024**|0.579 ± 0.017|0.558 ± 0.020|
>
> Table `aUp9.3`: Ablation results on normalization and activation layers with REVE-Small
>
> Our results show RMSNorm and GEGLU yield the best average performance. RMSNorm also helped stabilize training, consistent with findings in the original paper.
>
> ## Citation formatting
>
> We thank the reviewer for the helpful style suggestion. While the NeurIPS guidelines offer liberty regarding the formatting of citations, we agree that differentiating parenthetical citation and narrative citation improves the overall readability. The final manuscript will include the suggested modifications.
>
> ## References
>
> - Wang, Jiquan, Sha Zhao, Zhiling Luo, Yangxuan Zhou, Haiteng Jiang, Shĳian Li, Tao Li, Gang Pan. "CBraMod: A Criss-Cross Brain Foundation Model for EEG Decoding." The Thirteenth International Conference on Learning Representations. 2025.
> - Chowdhery, Aakanksha, et al. "Palm: Scaling language modeling with pathways." Journal of Machine Learning Research 24.240 (2023): 1-113.
> - Zhang, Biao, and Rico Sennrich. "Root mean square layer normalization." Advances in neural information processing systems 32 (2019).

---

> > ### Comment · Reviewer_aUp9 · 2025-08-03
> >
> > Thank you for your comprehensive rebuttal and additional experiments. The authors have adequately addressed most of my concerns regarding linear probing results, architectural clarifications, and ablation studies. I maintain my positive score and look forward to the open-sourced code and pretrained weights contributing to and inspiring the EEG research community.

---

> > > ### Author Response · Authors · 2025-08-07
> > >
> > > Dear reviewer `aUp9`,
> > >
> > > We appreciate your response and are glad to hear that your main concerns have been addressed. We are looking forward to sharing our model and pretrained weights with the community. Please don’t hesitate to reach out if you have any further questions about our work.

---

### Official Review · Reviewer_G86N · 2025-07-02

**Clarity:** 3
**Significance:** 3
**Originality:** 3
**Rating:** 4
**Confidence:** 3

**Summary:**

The paper introduces REVE (Representation for EEG with Versatile Embeddings), a foundation model for electroencephalogram (EEG) designed to address the heterogeneity of EEG datasets by enabling generalization across diverse recording setups, electrode configurations, and task types. The core challenge addressed is the limited generalizability of existing EEG foundation models, which often rely on homogeneous datasets and fixed positional encodings. Key Contributions:

Novel 4D Positional Encoding: REVE uses a Fourier-based 4D encoding that integrates spatial (3D electrode coordinates) and temporal (patch indices) dimensions, allowing flexibility in processing signals of arbitrary length and electrode arrangements. This addresses the rigidity of prior absolute or convolutional encodings.

Large-Scale Heterogeneous Pretraining: REVE is pretrained on 60,000+ hours of EEG data from 92 datasets spanning 25,000 subjects, the largest such effort to date. This diversity enhances generalization to unseen setups.

Spatio-Temporal Block Masking: A joint masking strategy disrupts contiguous regions in both spatial and temporal dimensions during pretraining, improving representation learning for robust feature extraction.

State-of-the-Art Performance: REVE achieves superior results on 10 downstream tasks, including motor imagery classification, seizure detection, and sleep staging, outperforming prior models like CBraMod and LaBraM.

**Questions:**

Question1: How does the 4D encoding perform with extremely sparse electrode setups (e.g., <4 channels) or non-standard montages not seen in pretraining? Are there limitations in capturing fine-grained spatial patterns with low channel counts?


Question2: The pretraining corpus includes clinical, BCI, and cognitive data, but how is data quality ensured? Are there biases in dataset selection (e.g., overrepresentation of certain demographics)?

Question3: How does REVE perform in zero-shot or few-shot scenarios, where minimal task-specific data is available? The current evaluation focuses on fine-tuning, but real-world applications often have limited labels.


Question4: Does the 4D encoding incur higher computational costs than prior methods? How does it scale with increasing channel counts and signal length?

**Ethical Concerns:**

["Major Concern: Improper research involving human subjects"]

**Limitations:**

yes

**Quality:**

3

**Strengths And Weaknesses:**

Strengths

Quality:

The experimental design is rigorous, with comprehensive evaluations across diverse tasks and datasets. The use of linear probing and fine-tuning protocols ensures fairness against baselines.
The large pretraining corpus and clear ablation studies (e.g., masking ratio, secondary loss impact) validate the method’s robustness.

Clarity:

The paper is well-structured, with detailed explanations of the 4D encoding, masking strategy, and training pipeline. Technical details (e.g., hyperparameters, dataset composition) are provided in appendices.
Visualizations (e.g., pretraining framework diagram) and tables (performance metrics) aid comprehension.

Significance:

REVE advances EEG foundation modeling by addressing long-standing challenges in data heterogeneity, which could accelerate progress in brain-computer interfaces (BCIs) and clinical diagnostics.
The focus on generalizable representations reduces reliance on task-specific fine-tuning, making it accessible for low-data scenarios.

Originality:

The 4D positional encoding is a novel approach for EEG, enabling true cross-setup generalization. The scale of pretraining and integration of spatial-temporal masking are also innovative.

Weaknesses

Quality:

While the dataset is large, the preprocessing (e.g., z-score normalization, clipping) might introduce biases. The impact of data cleaning on downstream tasks is not fully explored.
Some ablation results (e.g., Table 18) show inconsistent performance across masking ratios, leaving questions about optimal configurations for specific tasks.

Clarity:

The 4D encoding’s mathematical derivation (Section 2.2) could be more transparent, especially regarding how Fourier frequencies are selected and truncated to match embedding dimensions.
The secondary loss function’s design (attention pooling across layers) is briefly described; more intuition on its effectiveness in preventing over-specialization is needed.

Significance:

The clinical impact is implied but not directly validated (e.g., case studies in real-world diagnostics). The model’s utility in resource-constrained settings remains unaddressed.

Originality:

While the 4D encoding is novel, the use of masked autoencoding (MAE) follows prior work in vision. The contribution lies more in adaptation than radical architectural innovation.

---

> ### Author Rebuttal · Authors · 2025-07-31
>
> We thank Reviewer `G86N` for their feedback, highlighting the novelty of our 4D positional encoding, our strong experimental design, and the clarity of our work. Below, we clarify and directly address each of the concerns raised regarding the positional encoding, the secondary loss, data preprocessing, demographic biases, ethical concerns, practical applicability, and task specific configurations.
>
> ## 4D positional encoding
>
> The results and discussion of this part will be integrated into the final manuscript.
>
> > The 4D encoding’s mathematical derivation (Section 2.2) could be more transparent
>
> If the embedding size does not match $2\cdot n_{freq}^4$, we round up, compute the full 4D Fourier basis, and truncate as needed. For released checkpoints (base: 512, large: 1250), $n_{freq}$ is chosen to match exactly, so truncation isn’t required.
> > Does the 4D encoding incur higher computational costs than prior methods? How does it scale with increasing channel counts and signal length?
>
> The 4D encoding adds minimal compute overhead, with sinusoidal computations and a small linear layer. Cost scales linearly with the number of input tokens (channels × temporal patches) and is negligible relative to the transformer backbone.
> > How does the 4D encoding perform with extremely sparse electrode setups (e.g., <4 channels) or non-standard montages not seen in pretraining? Are there limitations in capturing fine-grained spatial patterns with low channel counts?
>
> The TUEV downstream task uses a bipolar montage, covering the case of montages not seen during pretraining.
> For sparse input, we benchmarked REVE-Base on MI (Left-Right) and imagined speech tasks while reducing channels by relevancy in line with Panachakel et al. (2021) and Abdullah et al. (2022):
>
> |Channels|64|32|16|8|4|2|1|
> |-|-|-|-|-|-|-|-|
> |PhysionetMI L-R|0.824 ± 0.008|0.808 ± 0.007|0.787 ± 0.008|0.781 ± 0.006|0.728 ± 0.009|0.700 ± 0.025|0.660 ±0.019|
> |Speech|0.565 ±0.016|0.490 ± 0.094|0.469 ±0.014|0.294 ± 0.063|0.258 ± 0.019|0.228 ± 0.006|0.209 ± 0.008|
>
> Table `G86N.1`: Sparse input ablation results on Fine-Tuning REVE-Base
>
> The MI performance goes from 0.824 with 64 channels to 0.660 with a single one, showing robustness to sparse input.
> For the imagined speech task, it goes from 0.565 to 0.258 with four channels and 0.209 with one, barely above the random-chance level, highlighting that REVE’s flexibility has limits when broad spatial coverage is required.
>
> ## Secondary loss intuition
>
> > The secondary loss function’s design (attention pooling across layers) is briefly described; more intuition on its effectiveness in preventing over-specialization is needed.
>
> To provide more intuition, as described in the paper:
> “This secondary loss encourages the encoder to distribute useful information across all layers, mitigating over-specialization in the final layer and yielding more generalizable representations.”
> The secondary objective reconstructs masked tokens using a compressed, global representation from attention pooling. This pooling acts as an information bottleneck, forcing the model to distill key information from the entire input sequence into a single vector. As shown by table 17 and its extension with results following Table aUp9.2, the secondary loss mainly improves the quality of the frozen embeddings of the model. We agree that this discussion would have been beneficial for the reader, and therefore changed the text accordingly.
>
> ## Data preprocessing
>
> Our pipeline applies z-score normalization followed by clipping at 15 standard deviations. This choice was motivated by stability concerns when training on large and heterogeneous datasets, as highlighted in Appendix A.2 of Défossez et al. (2023), which notes the harmful impact of extreme outliers on the training run: “Outliers can also cause extreme gradients and throw off the optimization process.”
>
> We acknowledge that our pipeline may limit exploration of more elaborate preprocessing or artifact mitigation strategies. However, given the scale and compute cost of pretraining REVE, we prioritized a streamlined and robust approach. We will update the manuscript to clarify this rationale.
>
> ## Demographic Bias
>
> > The pretraining corpus includes clinical, BCI, and cognitive data, but how is data quality ensured? Are there biases in dataset selection (e.g., overrepresentation of certain demographics)?
>
> Demographic bias is indeed a key concern when training foundation models at scale. Our pretraining corpus aggregates 92 publicly available EEG datasets spanning over 25,000 subjects. While this reduces overfitting to any single source, most public EEG data originates from North America and Europe, resulting in limited demographic diversity—an issue we recognize as a key limitation and a priority for the field to address.
>
> To further mitigate demographic and dataset imbalances, we leverage self-supervised learning (MAE). As shown by Xu et al. (2023), MAE is robust to long-tailed and imbalanced data, making it a practical choice for heterogeneous EEG corpora where some populations or recording conditions are underrepresented. While large-scale, diverse pretraining helps, equitable data collection remains an open challenge for future EEG foundation models.
>
> This additional information will be added to the final manuscript.
>
> ## Ethical concerns
>
> > Ethical Concerns: Major Concern: Improper research involving human subjects
>
> We recognize the importance of ethical considerations in research involving human subjects. However, as indicated in the NeurIPS checklist, this concern does not apply to our work. We exclusively used publicly available, open-access datasets that have already undergone review and approval by relevant scientific and ethics committees. No new data were collected, and no direct interaction with human subjects took place in the course of our study.
>
> ## Practical clinical impact and few shot performance
>
> > The clinical impact is implied but not directly validated (e.g., case studies in real-world diagnostics). The model’s utility in resource-constrained settings remains unaddressed.
> > How does REVE perform in zero-shot or few-shot scenarios, where minimal task-specific data is available? The current evaluation focuses on fine-tuning, but real-world applications often have limited labels.
>
> We conducted experiments in a few-shot (FS) setting, to simulate the use of a BCI system. We use the BCI IV-2a dataset to construct FS tasks with 2 classes (Left-Right). For each subject, we perform multiple inductive FS runs. In each run, we randomly select N labeled samples per class (“shots”) within a session and use the remaining samples from both sessions to evaluate performance. We classify using a Nearest Class Mean method, and repeat the runs 20 times per subject. We report the average balanced accuracy across subjects and runs.
>
> Below the results with REVE-Base.
>
> |N-shots|1|2|5|10|20|
> |-|-|-|-|-|-|
> |REVE-Base (only SSL)|0.588 ± 1.45|0.601 ± 0.001|0.652 ± 0.013|0.688 ± 0.010|0.723 ± 0.010|
> |REVE-Base + second pretraining|0.605 ± 1.12|0.645 ± 0.009|0.705 ± 0.009|0.768 ± 0.009|0.817 ± 0.004|
>
> Table `G86N.2`: **Few-shot** performance of REVE-Base on BCI IV-2a dataset
>
> The first row of the table shows the model's performance directly after SSL pre-training, without further adaptation. REVE's embeddings can be leveraged even without supervised fine-tuning or probing, highlighting the potential for direct application to BCI tasks. This level of generalization is rare among BCI embedding models, as most existing approaches require task-specific retraining (e.g., from other subjects in the dataset).
>
> In the second row, we see the result of a second experiment, involving supervised cross-dataset (MI LR) training on REVE using Left-Right MI open datasets (Schirrmeister 2017, Cho 2017, Physionet MI, Lee 2019, Weibo 2014, and Large). REVE's 4D positional encoding allows the joint pre-training across all electrode configurations without channel alignment or selection typically required in such scenarios. This step improved FS performance across all shot configurations, achieving gains of up to 10% at 20 shots. A summary table of these results will be included in the final manuscript's appendices.
>
> Proper zero-shot with REVE would require a second modality (e.g. text, image) to anchor label semantics. Nonetheless, REVE’s high-quality representations enable flexible plug-in classifiers, which is a practical direction for EEG where labeled data is scarce.
>
> ## Optimal task specific configurations
>
> We acknowledge that the ablation on the masking ratios, based Table 18, could be more impactful if we include more results confirming that an optimal setting exists for the majority of the tasks. We thus ran those ablations on 8 DT in order to report the average below.
>
> |Masking Ratio|Frozen Random|Frozen Block|Full Fine-Tuning Random|Full Fine-Tuning Block|
> |-|-|-|-|-|
> |0.25|0.523|0.513|0.612|0.602|
> |0.55|0.550|**0.558**|0.643|**0.665**|
> |0.75|0.519|0.546|0.606|0.655|
>
> Table `G86N.3`: Average ablation results on masking ratios with REVE-Small
>
> The results comfort our choice of a masking ratio of 0.55, we will consequently update table 18 in the final manuscript, with the 8 tasks.
>
> ## References
>
> - Panachakel, Jerrin Thomas, and Angarai Ganesan Ramakrishnan. "Decoding covert speech from EEG-a comprehensive review." Frontiers in neuroscience 15 (2021)
> - Abdullah, Ibrahima Faye, and Md Rafiqul Islam. "EEG channel selection techniques in motor imagery applications: A review and new perspectives." Bioengineering (2022)
> - Défossez, Alexandre, et al. "Decoding speech perception from non-invasive brain recordings." Nature Machine Intelligence 5.10 (2023)
> - Xu, Zhengzhuo, Ruikang, Liu, Shuo, Yang, Zenghao, Chai, Chun, Yuan. "Learning imbalanced data with vision transformers." CVPR 2023

---

> > ### Comment · Reviewer_G86N · 2025-08-04
> >
> > Thanks for your responses. My main concerns are addressed, so I decide to keep first rate.

---

> > > ### Author Response · Authors · 2025-08-07
> > >
> > > Dear reviewer `G86N`,
> > >
> > > Thank you for your response and for letting us know that your main concerns have been addressed.
> > > If you have any other questions, feel free to reach out.

---

### Official Review · Reviewer_rHPq · 2025-07-02

**Clarity:** 3
**Significance:** 2
**Originality:** 3
**Rating:** 3
**Confidence:** 4

**Summary:**

REVE is a foundation model for EEG data that generalizes across diverse setups. It uses 4D positional encoding and was pretrained on 60,000+ hours of EEG data from 92 datasets (25,000 subjects). The model achieves SOTA performance on 10 tasks, including motor imagery, seizure detection, and emotion recognition.

**Questions:**

1. The introduction (lines 37-39) criticizes existing models for issues such as being trained solely on the TUH dataset, leading to poor generalization to EEG data with different electrode layouts. However, among the pretrained models cited in lines 35-36, none actually suffer from this limitation—for example, CBraMod, despite being trained only on TUH dataset, has been successfully applied to downstream tasks with varying electrode configurations. Could the authors clarify the basis for this claim?
2. Did the pretrained baseline models use the same two-stage fine-tuning strategy as the proposed method in this paper? If the results were directly taken from the original baseline papers without adopting the two-stage fine-tuning strategy, such a comparison might be unfair.
3. As mentioned in the weaknesses, could the author provide more results of the linear probing experiment to prove the performance of REVE under lightweight finetuning?

**Ethical Concerns:**

["NO or VERY MINOR ethics concerns only"]

**Limitations:**

Yes

**Quality:**

3

**Strengths And Weaknesses:**

Strengths: The paper demonstrates outstanding model performance and achieves state-of-the-art results in 10 different downstream EEG tasks. The paper provides particularly detailed experimental methodology, including thorough implementation specifics that ensure transparency. Importantly, the author ensured the repeatability of the model by publicly releasing all the code and pre-trained weights.
Weaknesses: While the paper emphasizes REVE’s advantages in linear probing and limited fine-tuning scenarios, its core experimental design exhibits notable gaps. Specifically:
1.	Abstract (lines 5-7) critiques existing EEG foundation models for their poor linear probing performance, and (lines 15-16) claim REVE’s strong generalization with “little to no fine-tuning.”
2.	The introduction (lines 41-42) further argues that current models require “full fine-tuning” for transfer.
3.	Results and Discussion (lines 319-322) highlight REVE’s “key advantage” in “producing high-quality latent spaces without heavy fine-tuning”.
Yet, the primary results (Table 2) rely on a two-stage fine-tuning strategy (Section 3.3 Downstream Tasks). Only limited validation for the above statement is provided (“w/PT” and “Frozen” settings in Table 3), tested on just one dataset, failing to generalize the “lightweight fine-tuning” advantage.
This inconsistency weakens the paper’s argument for REVE’s “lightweight fine-tuning benefits”, potentially undermining the conclusions’ credibility. To substantiate this argument, comparisons of all pre-trained baselines under linear probing across all downstream datasets are essential—a gap the current experiments leave unaddressed. Such validation is critical: if pre-trained models still require two full fine-tuning stages despite their pre-training, REVE’s position in the performance-simplicity tradeoff becomes significantly less competitive.

---

> ### Author Rebuttal · Authors · 2025-07-31
>
> We thank Reviewer `rHPq` for recognizing the strong performance of REVE across a diverse set of EEG tasks, as well as our commitment to reproducibility through public release of code and pre-trained models. We also thank the reviewer for their constructive and detailed feedback. Below, we address the key concerns regarding linear probing, the fine-tuning setup, and generalization to variable electrode layouts.
>
>
> ## Linear probing comparisons against baselines
>
>
> > could the author provide more results of the linear probing experiment to prove the performance of REVE under lightweight finetuning?
>
>
> We agree that our claims about REVE’s effectiveness under limited adaptation regimes could benefit from more empirical evidence. To address this, we have added linear probing results for CBraMod across our downstream tasks suite. Given the added clarity these results provide and following Reviewer `aUp9`’s suggestion, the complete table will be moved to the main paper.
>
> In more detail, we extended the results we had in Table 16. To ensure a fair comparison, we reproduced CBraMod (Wang et al., 2025) using their official code and pretrained checkpoint, carefully following their classification pipeline (notably, no pooling) and matched architectural details to avoid any bias. This resulted in performance gains for REVE of up to 16% over what was originally reported.
>
> All results are shown in the following table, where we present balanced accuracies.
>
> |Dataset|REVE-B (Pool)|REVE-B (No Pool)|REVE-L (Pool)|REVE-L (No Pool)|CBraMod (Pool)|CBraMod (No Pool)|
> |-|-|-|-|-|-|-|
> |Mumtaz|0.962 ± 0.003|0.931 ± 0.021|**0.985 ± 0.006**|0.980 ± 0.009|0.859 ± 0.009|0.907 ± 0.027|
> |Mental Arithmetic|0.725 ± 0.010|**0.740 ± 0.073**|0.712 ± 0.008|0.665 ± 0.103|0.500 ± 0.000|0.605 ± 0.020|
> |TUAB|0.810 ± 0.007|0.809 ± 0.004|**0.821 ± 0.004**|0.809 ± 0.004|0.500 ± 0.000|0.500± 0.000|
> |PhysioNetMI|0.537 ± 0.005|0.510 ± 0.012|0.551 ±0.001|**0.617 ± 0.000**|0.256 ± 0.002|0.531 ± 0.015|
> |BCIC-IV-2a|0.432 ± 0.004|0.517 ± 0.015|0.534 ±0.001|**0.603 ± 0.011**|0.287±0.023|0.376 ± 0.006|
> |ISRUC|0.697 ± 0.011|0.662 ± 0.030|0.743 ± 0.004|**0.758 ± 0.001**|0.407 ± 0.049|0.430 ± 0.043|
> |HMC|0.647 ± 0.008|0.604 ± 0.008|0.703 ± 0.003|**0.710 ± 0.007**|0.368 ± 0.001|0.538 ± 0.009|
> |BCIC2020-3|0.234 ± 0.009|**0.390 ± 0.017**|0.274 ± 0.001|0.378 ± 0.021|0.214 ± 0.003|0.374 ± 0.007|
> |TUEV|0.592 ± 0.008|0.508 ± 0.073|**0.630 ± 0.003**|0.550 ± 0.014|0.219 ± 0.009|0.482 ± 0.037|
> |Faced|0.240 ± 0.010|0.422 ± 0.028|0.283 ± 0.003|**0.469 ± 0.007**|0.117 ± 0.005|0.261 ± 0.013|
> |Avg.|0.586|0.609|0.623|**0.654**|0.373|0.501|
>
> Table `rHPq.1`: **Linear probing** results on downstream tasks for REVE and CBraMod. Best results are highlighted in bold.
>
> Note: To ensure exhaustiveness, we have added the FACED dataset to the linear probing experiments, which affects the reported averages.
> As shown, REVE consistently outperforms CBraMod across all downstream tasks under linear probing, regardless of whether pooling is used.
>
> ## On the “two stages” of fine-tuning
>
> > Did the pretrained baseline models use the same two-stage fine-tuning strategy as the proposed method in this paper? If the results were directly taken from the original baseline papers without adopting the two-stage fine-tuning strategy, such a comparison might be unfair.
>
> We agree that simplicity is critical to the practicality of foundation models and appreciate the concern regarding potential complexity in our fine-tuning procedure. To clarify: our “two stages” of fine-tuning are implemented as a single continuous training run, where the backbone is initially frozen (i.e., only the head is trained) and later unfrozen. This helps stabilize training but it is not a two-stage pipeline in the sense of full fine-tuning twice.
>
> CBraMod adopts a similar strategy via a multi-learning-rate setup (i.e., lower LR for the backbone), which serves a comparable purpose. Such strategies are common in transfer learning (Kumar et al., 2022). We will revise the paper to explicitly clarify this point and avoid any misinterpretation about the complexity or practicality of our fine-tuning approach.
>
> ## On generalization to varied electrode setups of other methods
>
> > The introduction (lines 37-39) criticizes existing models for issues such as being trained solely on the TUH dataset, leading to poor generalization to EEG data with different electrode layouts. However, among the pretrained models cited in lines 35-36, none actually suffer from this limitation—for example, CBraMod, despite being trained only on TUH dataset, has been successfully applied to downstream tasks with varying electrode configurations. Could the authors clarify the basis for this claim?
>
> Our introduction primarily refers to LaBraM (Jiang et al., 2024) and BIOT (Yang et al., 2024), which rely on fixed positional embeddings, making direct transfer to unseen electrode layouts infeasible (e.g., BIOT’s checkpoints use 16 or 18 fixed electrodes). We agree that CBraMod, through its convolution-based asymmetric conditional positional encoding (ACPE), can technically handle arbitrary layouts, but it requires training the weights of this module to reach the best performance.
> CBraMod is pretrained on a single electrode setup, as both number and order of electrodes must match. Adapting to a different configuration requires retraining / fine-tuning, so true cross-configuration transfer is not supported, unlike REVE.
>
> This is reflected in linear probing performance: as noted in the original CBraMod paper (Appendix J), “fixing the pre-trained parameters during training on downstream datasets will lead to a very large performance decline... CBraMod cannot currently serve as a fixed-parameter feature extractor like CLIP and SAM, and fine-tuning is still necessary.”.
>
> Our new linear probing results further confirm that these models struggle to generalize across electrode setups. We nonetheless acknowledge that the phrasing in our introduction could be clarified and will revise it to more accurately distinguish between the architectural limitations of LaBraM/BIOT and the more flexible design of CBraMod.
>
> ## References
>
> - Wang, Jiquan, Sha Zhao, Zhiling Luo, Yangxuan Zhou, Haiteng Jiang, Shĳian Li, Tao Li, Gang Pan. "CBraMod: A Criss-Cross Brain Foundation Model for EEG Decoding." The Thirteenth International Conference on Learning Representations. 2025.
> - Kumar, Ananya, et al. (2022) "Fine-Tuning can Distort Pretrained Features and Underperform Out-of-Distribution." International Conference on Learning Representations.
> - Jiang, Weibang and Liming Zhao, and Bao-liang Lu. "Large Brain Model for Learning Generic Representations with Tremendous EEG Data in BCI." The Twelfth International Conference on Learning Representations.
> - Yang, Chaoqi, M. Westover, and Jimeng Sun. "Biot: Biosignal transformer for cross-data learning in the wild." Advances in Neural Information Processing Systems 36 (2023): 78240-78260.

---

> ### Author Response · Authors · 2025-08-07
>
> Dear reviewer `rHPq`,
>
> As the discussion period is coming to an end, please feel free to reach out if you have any questions or if there is anything we can clarify further. We would also appreciate knowing if our rebuttal addressed your concerns.
>
> Thank you for your time and consideration.

---

### Official Review · Reviewer_xtx1 · 2025-07-03

**Clarity:** 4
**Significance:** 4
**Originality:** 2
**Rating:** 5
**Confidence:** 3

**Summary:**

This paper presents REVE a foundation model trained on over 60,000 hours of EEG data from 92 datasets and 25,000 subjects. REVE achieves SOTA performance on 10 downstream EEG tasks. REVE addresses several challenges in training an EEG foundation model such as data heterogenity, low SNR, and different electrode configurations. The main contributions of this paper are

1. A novel 4D positional encoding to model EEG signals with varying temporal lengths and electrode configuration
2. Curation of large scale EEG by combining 92 datasets.
3. State-of-the-art performance on multiple benchmarks

**Questions:**

What other architecture design choices did you consider? Were there any other successful or failed designs that are not mentioned in the paper. It would be great to mention failed designs as that can help guide future directions

Did you have an ablation on performance vs scale to find out whether there is room for improvement with more data or we have reached a saturation in performance with this dataset

**Ethical Concerns:**

["NO or VERY MINOR ethics concerns only"]

**Final Justification:**

The authors addressed most of my concerns. I will maintain positive rating

**Limitations:**

yes

**Quality:**

4

**Strengths And Weaknesses:**

1. This paper has created a large scale EEG dataset by combining 92 datasets spanning data from clinical, BCI and research domains. This will be a valuable resource for foundation model research in EEG
2. This paper adapts MAE to spatiotemporal EEG signal and proposes a 4D positional encoding to address variation in temporal length and electrode configuration across datasets.
3. Strong results demonstrating the importance of large scale pretraining on multiple downstream tasks.
4. Clear writing , description of methods and results.
5. Modeling choices are inspired by successes in different modalities

I do not see any major weaknesses in this paper and believe it is a strong contribution towards foundation model development in EEG Domain. A minor weakness of this paper is that it is leveraging the ideas from other modalities but is missing an ablation study on the design choices for the architecture.

---

> ### Author Rebuttal · Authors · 2025-07-31
>
> We thank Reviewer `xtx1` for their thoughtful review of our contributions. We appreciate the recognition of the novelty of our 4D positional encoding, the significance of our large-scale dataset, our methodological choices, and the impact of our results.
>
>
> > A minor weakness of this paper is that it is leveraging the ideas from other modalities but is missing an ablation study on the design choices for the architecture.
>
>
> We acknowledge that we did not explore many architecture variations, as this was not the primary focus of our study. Our design choices were mainly guided by prior work across modalities, as discussed in Section 2.3. Transformers have demonstrated strong, modality-agnostic performance, with their core architecture largely unchanged across vision or text domains.
> Nevertheless, as reported in Table `aUp9.2` (our answer to reviewer `aUp9`), we provide additional ablation studies on normalization and activation layers. These results support our design choices, as our selected architecture consistently achieved the best performance which aligns well with best practices in modern Transformer models for language.
>
>
> We report here the average performance across the downstream tasks, see Table `aUp9.1` for the exhaustive results.
>
>
> |Dataset|GEGLU + RMSNorm|GELU + RMSNorm|GEGLU + LayerNorm|
> |-|-|-|-|
> |Avg.|**0.596 ± 0.024**|0.579 ± 0.017|0.558 ± 0.020|
>
>
> Table `xtx1.1`: Average of the ablation results on normalization and activation layers
>
>
> > What other architecture design choices did you consider? Were there any other successful or failed designs that are not mentioned in the paper. It would be great to mention failed designs as that can help guide future directions
>
> To leverage the Transformer architecture to EEG data, our experimentation focused on patching strategy, positional encoding, and training objectives rather than altering the backbone.
>
>
> For positional encoding, we initially tested separate spatial and temporal 1D Fourier encodings, but a unified approach proved more practical. We also obtained competitive results with 4D learnable positional embeddings, but this approach limited our ability to generalize to unseen spatial positions and timesteps.
>
>
> For the patching strategy, we explored various patch sizes and overlaps. The optimal configuration we found was very similar to what has already been reported in the LaBraM (Jiang et al., 2024), BIOT (Yang et al., 2024) and CBraMod (Wang et al., 2025) baselines.
>
>
> For self-supervised learning, we experimented with random masking instead of block masking, as well as different masking ratios. Following reviewer G86N comments, we ran additional ablations on masking strategy to extend results presented in the Table 18 of the appendix E of the original manuscript.
>
>
> > Did you have an ablation on performance vs scale to find out whether there is room for improvement with more data or we have reached a saturation in performance with this dataset
>
>
> We thank the reviewer for mentioning the topic of data scaling law. We did observe hints of scaling effects, but setting up robust scaling experiments was challenging due to the need for extensive data curation, which often requires a pretrained model. As a result, our scaling attempts remained limited by the inherent noise in EEG data. In practice, data curation is usually pursued after a foundation model is established, and we see this as a key direction for future work.
>
>
> ## References
>
>
> - Jiang, Weibang and Liming Zhao, and Bao-liang Lu. "Large Brain Model for Learning Generic Representations with Tremendous EEG Data in BCI." The Twelfth International Conference on Learning Representations.
> - Yang, Chaoqi, M. Westover, and Jimeng Sun. "Biot: Biosignal transformer for cross-data learning in the wild." Advances in Neural Information Processing Systems 36 (2023): 78240-78260.
> - Wang, Jiquan, Sha Zhao, Zhiling Luo, Yangxuan Zhou, Haiteng Jiang, Shĳian Li, Tao Li, Gang Pan. "CBraMod: A Criss-Cross Brain Foundation Model for EEG Decoding." The Thirteenth International Conference on Learning Representations. 2025.

---

> ### Author Response · Authors · 2025-08-07
>
> Dear reviewer `xtx1`,
>
> As the discussion period is coming to an end, please let us know if you have any further comments or questions. We are happy to address any remaining concerns or provide clarifications if needed.

---

### Note · Authors · 2025-08-14

We thank the reviewers for their constructive feedback, and have exhaustively addressed all questions and concerns, which has helped us to significantly strengthen the paper. Below is a concise summary of our rebuttal and revisions.

## Additional experiments

We extended **linear probing** to all downstream tasks, replicated CBraMod with official code/checkpoint under **matched pipelines** (with and without pooling), and confirmed that REVE consistently outperforms baselines with a frozen encoder. Additional ablations now cover RMSNorm vs. LayerNorm, GEGLU vs. GELU, masking ratios, and secondary loss settings across a wider DT range. We evaluated robustness to **sparse channels** and **non-standard montages** (bipolar), noting limits when dense spatial coverage is needed. On **few-shot MI** (BCI IV-2a), a nearest-class-mean classifier performs strongly without fine-tuning, with further gains after cross-dataset supervised adaptation, demonstrating the quality of REVE’s embeddings and the role of 4D encoding in enabling cross-dataset training without matched electrode configurations.

## Clarifications

We confirmed that 4D positional encoding adds **negligible compute overhead**, and we **reported full training cost**. We strengthened evidence for lightweight adaptation via matched, fair comparisons, clarified that the **“two stages” are a single run** with delayed unfreezing, and refined our **discussion of generalization** across electrode layouts, highlighting differences between LaBraM/BIOT and CBraMod. We explained **preprocessing choices** (z-score + clipping) for stability at scale, and acknowledged **demographic imbalances and data quality** as limitations. Ethics/privacy notes now emphasize research-only use, the need for further validation before clinical deployment, non-release of the decoder, and a commitment to retrain if dataset accessibility changes.

## Manuscript revisions

Key changes include moving the full linear-probing table to the main text, expanding ablations, adding summaries of sparse-channel and few-shot studies, noting the negligible cost of 4D encoding and reporting training cost, clarifying delayed unfreezing and architecture terminology, refining prior work statements, adding follow-up research directions in the ethics section, and improving citations and formatting.

---

### Decision · Program_Chairs · 2025-09-17

**Decision:**

Accept (poster)

**Comment:**

This paper presents REVE, the largest EEG foundation model to date, trained on 60,000+ hours from 92 datasets and 25,000 subjects. Its key innovation is a 4D positional encoding that enables generalization across variable electrode layouts and signal lengths. REVE achieves state-of-the-art results on 10 diverse EEG tasks with strong performance under both fine-tuning and lightweight adaptation.

The paper is well-executed and impactful, with comprehensive evaluations, meaningful ablations, and a strong commitment to reproducibility through open code and pretrained weights. While interpretability and clinical validation remain areas for future work, the contribution is substantial, addressing long-standing challenges in EEG modeling.